

# Modeling on the drought stress impact on the summertime biogenic isoprene emissions in South Korea

Yong-Cheol Jeong[1], Yuxuan Wang[1], Wei Li[1,2], Hyeonmin Kim[3], Rokjin J. Park[3], Mahmoudreza Momeni[1]

[1]Department of Earth and Atmospheric Sciences, University of Houston, Houston, TX, USA
[2]Now at National Oceanic and Atmospheric Administration (NOAA) Air Resource Laboratory
[3]School of Earth and Environmental Sciences, Seoul National University, Seoul, South Korea

*Correspondence to*: Yuxuan Wang (ywang246@central.uh.edu)

**Abstract.** Biogenic isoprene emissions play an important role in air quality so it is important to quantify their response to extreme events such as drought. While there have been some efforts to reduce uncertainties in isoprene emissions under the
drought conditions, the effort has not been made in the South Korean region. Here, we aimed to constrain drought stress on biogenic isoprene emissions in South Korea using satellite formaldehyde (HCHO) column, the key product of isoprene oxidation, and a chemistry transport model (GEOS-Chem) with Model of Emissions of Gases and Aerosols from Nature version 2.1 (MEGAN2.1). It was found that the HCHO column from the Ozone Monitoring Instrument (OMI) increased by 5.4 % under the drought condition compared to the normal condition, but GEOS-Chem simulated a 20.23 % increase indicating
an overestimation of isoprene emissions under drought. We implemented two existing drought stress algorithms in MEGAN2.1 and found they were not effective to reduce HCHO column biases in South Korea because those algorithms were proposed and developed for the Southeast United States (SE US). To improve this, we applied an iterative finite difference mass balance (IFDMB) method to estimate isoprene emissions using the OMI HCHO column. With this method, isoprene emissions were reduced by 60 % under the drought conditions compared to those simulated by the standard MEGAN2.1 implemented in
GEOS-Chem. The increase of HCHO column under the drought conditions compared to the normal condition was also reduced to 10.71 %, which is comparable to that in the satellite retrievals. Based on isoprene emission difference between MEGAN2.1 and IFDMB, we developed the empirical equations to adjust isoprene emissions in South Korea that also improved model prediction of the secondary pollutant such as ozone.

## 1 Introduction

In the troposphere, biogenic non-methane volatile organic compounds (BVOCs) are important to regional air quality because they are precursors to tropospheric ozone and secondary organic aerosols (Atkinson, 2000; Holm and Balmes, 2022; Pacifico et al., 2009). BVOCs are emitted from terrestrial vegetations and 70 % of global BVOCs emissions are isoprene emissions (Pacifico et al., 2009; Sindelarova et al., 2014). Isoprene emissions depend on not only physiological factors such as plant



functional type, leaf area index, and leaf age, but also meteorological factors such as temperature, radiation, and soil moisture

(Guenther et al., 2012; Guenther et al., 2006). For example, high temperature can increase isoprene emissions by promoting isoprene synthases activity in the vegetations, while low soil moisture can modulate isoprene emissions by reducing stomatal conductance photosynthesis rates of terrestrial vegetations (Ferracci et al., 2020; Huang et al., 2015; Potosnak et al., 2014; Seco et al., 2022) Given this, it is of great importance in the air quality study to quantify such climatic or meteorological impacts on isoprene emissions.

Drought is one of the most important climatic or meteorological extreme events that can modulate isoprene emissions by high temperature and low soil moisture (Limousin et al., 2010; Zhou et al., 2014; Rissanen et al., 2022; Huang et al., 2015; Seco et al., 2015). Although the response of isoprene emissions is non-monotonic to the drought severity, isoprene emissions tend to increase under drought conditions in general (Ferracci et al., 2020; Funk et al., 2005; Pegoraro et al., 2004; Potosnak et al., 2014) while the extent of the increase is challenging to quantify because the number of in-situ isoprene measurements

worldwide is spatiotemporally scarce. Taking advantage of wide spatiotemporal coverage of satellite retrievals, some studies have used the tropospheric formaldehyde (HCHO) column retrievals from the satellite to estimate isoprene emissions response to drought. It is well known that HCHO could be used as a proxy for isoprene emissions because HCHO is produced fast with high yield from isoprene oxidation (Wolfe et al., 2016; Palmer et al., 2003; Sprengnether et al., 2002). Some previous studies showed that the tropospheric HCHO column from the Ozone Monitoring Instrument (OMI) on the Aura satellite increased by

6.5–22 % in the southeastern United States (US) region during the summertime drought, which is indicative of the increase of isoprene emissions during drought (Li et al., 2022; Naimark et al., 2021).

The Model of Emissions of Gases and Aerosols from Nature (MEGAN) (Guenther et al., 2006; Guenther et al., 2012) is widely used to estimate isoprene emissions as a function of physiological and meteorological factors. However, isoprene emissions simulated by MEGAN showed a large overestimation under the drought condition, so some efforts have been carried out in

various ways to reduce such bias (Wang et al., 2022a; Wang et al., 2021). In Jiang et al. (2018), for example, a drought stress algorithm for MEGAN isoprene emission was proposed based on observational constraints from isoprene flux measurements at the Missouri Ozarks AmeriFlux (MOFLUX) site in the southeastern US (SE US) during two summertime drought conditions (summers of 2011 and 2012). Wang et al. (2022b), on the other hand, used the OMI HCHO column as top-down constraints to derive a drought stress algorithm for MEGAN isoprene emissions in order to achieve a wider spatiotemporal coverage. They

compared the OMI HCHO column with the HCHO column simulated by a chemical transport model (GEOS-Chem) with MEGAN isoprene emission in the SE US region for a multiple-year period (2005–2017 summers) and showed that the model HCHO column had larger biases under the drought conditions compared to the normal/wet conditions. To reduce the model HCHO biases under the drought conditions, they derived a drought stress algorithm for MEGAN isoprene emission which





could minimize the HCHO biases in the SE US region. This algorithm reduced isoprene emission by 8.6–20.7 % in the
summertime drought condition in the SE US.

In South Korea, it is also known that isoprene emissions play an important role on air quality (Kim et al., 2018; Kim et al., 2013; Lee and Park, 2022). Observational evidence of the impact of drought stress on isoprene emissions (Wasti and Wang, 2022) showed that the OMI HCHO column increased by 2.97 % in the mild drought and by 8.02 % in the extreme drought in the summertime in the South Korean region, indicating the increase of isoprene emissions under the drought condition.
However, an effort to evaluate MEGAN isoprene emissions in South Korea has not been made yet despite the wide use of MEGAN in air quality simulations for South Korea (Jang et al., 2020; Lee and Park, 2022; Lee et al., 2014; Yu et al., 2023). Following this need, the present study aimed to constrain isoprene emissions under the summertime (June-July-August, JJA hereafter) drought condition in the South Korean region by using the chemical transport model (GEOS-Chem) with MEGAN and satellite HCHO columns. We first investigated the HCHO column biases of GEOS-Chem under the drought condition.
After we verified that the current two drought stress algorithms for MEGAN isoprene emissions were not effective to reduce the HCHO column biases in South Korea, we conducted a top-down inversion of isoprene emissions in this region and estimated the drought impact on isoprene emissions based on the inversion results.

## 2. Data and Method

### 2.1. Drought index

We adopted a gridded drought index, the evapotranspiration deficit index (DEDI), to detect the drought in South Korea region. The DEDI is derived from actual evapotranspiration (AET) and potential evapotranspiration (PET) datasets (Zhang et al., 2023), which are provided by the European Centre for Medium-Rand Weather Forecast (ECMWF) Reanalysis version 5 (ERA5) (Hersbach et al., 2020). Because the DEDI considers a balance between the atmospheric evaporation demand and the actual land water evaporated from soil, water surfaces and vegetation, the DEDI can connect the climate system with terrestrial
ecosystems (Zhang et al., 2023). The horizontal resolution of the DEDI data was $0.25° \times 0.25°$ and it was regridded to $0.25° \times 0.3125°$ to match with the horizontal resolution used in the following model simulations. Consistent with the previous studies (Wang et al., 2022b), we focused on the weekly drought by averaging daily DEDI values in all grid points. To decide the drought thresholds of the DEDI data in South Korean region, we followed the percentile category approach. As described in the previous studies (Zhang et al., 2023; Zhang et al., 2022), the 30 % percentile value (-0.49) of the climatological DEDI
(1981 – 2010; 30 years) was obtained to detect drought periods (DEDI ≤ -0.49). Figure S1 shows the weekly DEDI values in the recent seven summers (2016 – 2022 JJA) and compared them with the Standardized Precipitation Index 1 (SPI1) (Mckee et al., 1993) which is based on the 1-month accumulated precipitation measurements from the in-situ sites in South Korea. The daily SPI1, provided by the Korea Meteorological Administration (KMA), was averaged to derive the weekly SPI1, and the





DEDI value at the nearest point to each SPI1 in-situ site was used here for comparison with the SPI1. As described in Zhang
et al. (2023), DEDI had consistent drought conditions with the SPI1 (Fig. S1), indicating that DEDI can represent the observed
drought conditions. Both drought indices showed that 2016 JJA, 2017 JJA, and 2018 JJA had strong and persistent drought
conditions compared to other summers, so we focused on these three summers (2016 – 2018 JJA) in the following analyses
and model simulations.

## 2.2 In-situ observations of Ozone and PM$_{2.5}$ and the OMI HCHO product

We used the observed in-situ surface ozone (O$_3$) and particulate matter with a diameter less than or equal to 2.5μm (PM$_{2.5}$)
datasets in South Korea for 2016 – 2018 JJA period. The hourly O$_3$ (PM$_{2.5}$) concentrations at all available sites were used in
this study (Fig. 1) to derive daytime mean (7am – 6pm) O$_3$ (PM$_{2.5}$) concentrations in each day. We focused on daytime because
most of the biogenic isoprene emissions occur in daytime and BVOCs react with nitrogen oxides (NO$_x$) under the presence of
solar radiation to produce secondary pollutants, especially O$_3$ (Atkinson, 2000). The weekly daytime-mean O$_3$ (PM$_{2.5}$)
concentration at each site was derived by averaging daily daytime mean O$_3$ (PM$_{2.5}$) concentrations.

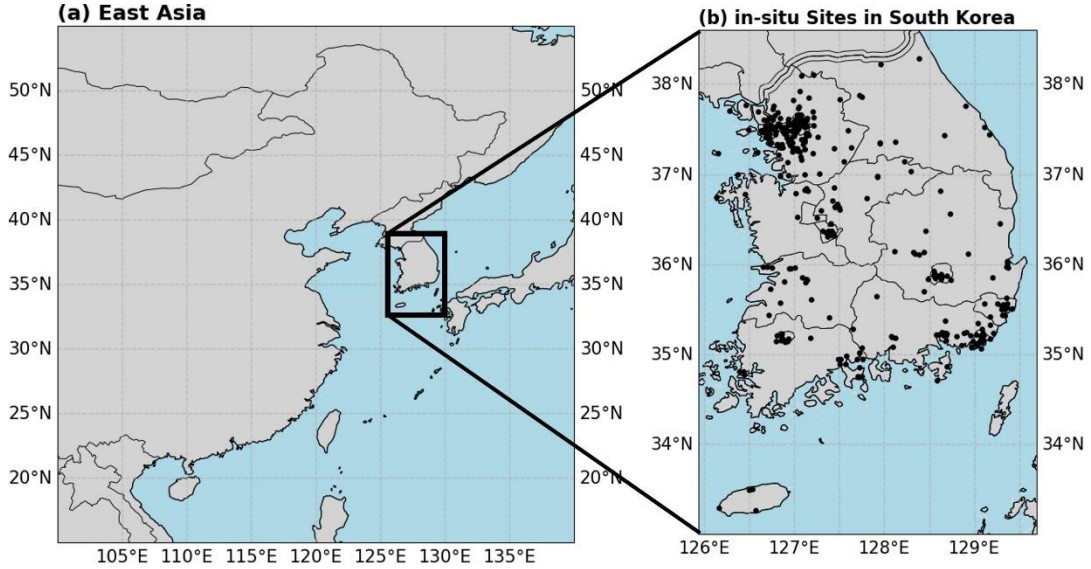

**Figure 1: (a) The location for South Korea in East Asia and (b) the locations for in-situ air pollutant measurements in South Korea.**

The Ozone Monitoring Instrument (OMI) v003 Level 3 tropospheric formaldehyde (HCHO) column density data (OMHCHOd)
were used in this study. All pixels with bad formaldehyde retrievals, high cloud fractions (>30 %), high solar zenith angle
(>70°), and pixels affected by OMI's row anomaly in Level 2 HCHO column data (OMHCHO) have been already filtered out



in OMHCHOd data (Chance, 2019). The horizontal resolution was 0.1° x 0.1°, and it was regridded to 0.25° × 0.3125° to match with the horizontal resolution used in the model simulations. Zhu et al. (2020) verified the OMI HCHO column retrievals with the HCHO aircraft observation in South Korea region during the Korea–United States Air Quality (KORUS-AQ)

campaign from the May to early June 2016. They found that the OMI HCHO column retrievals were biased low by about 22 % relative to the aircraft observation. Following the consistent method with previous studies (Shen et al., 2019; Wang et al., 2022b), we corrected this bias by applying factor of 1.28 (1 / (1–0.22)) to the OMI HCHO column retrievals.

### 2.3 GEOS-Chem model simulations

We used the atmospheric composition simulations of the nested-grid GEOS-Chem chemical transport model (version 12-7-2)

over East Asia (100°E–140°E, 15°N–55°N; Fig. 1) with a horizontal resolution of 0.25° × 0.3125° (Bey et al., 2001; Chen et al., 2009; Wang et al., 2004) to obtain the modeled isoprene emissions and HCHO columns in 2016 – 2018 JJA period. The model was driven by GEOS-forward processing (GEOS-FP) meteorological data from NASA's Global Modeling and Assimilation Office (GMAO). The chemical boundary conditions for the nested domain were obtained from the GEOS-Chem global simulations with a horizontal resolution of 2° × 2.5° every three hours. The GEOS-Chem Tropchem mechanism was

used for the gas-phase chemistry mechanisms in GEOS-Chem (Fisher et al., 2016; Mao et al., 2013; Marais et al., 2016) and gas-aerosol phase partitioning of the nitric acid and ammonia was calculated by ISORROPIA II (Fountoukis and Nenes, 2007). For anthropogenic emissions in East Asia region, we used the Korea-United States Air version 5 (KORUSv5) emission inventory (Woo et al., 2020), which is one of the most recent anthropogenic emission inventories in East Asia. The Global Fire Emissions Database 4 inventory (GFED4) (Van Der Werf et al., 2010) was used for biomass burning, and biogenic

emissions such as isoprene emissions were simulated by MEGAN version 2.1 (MEGAN2.1) (Guenther et al., 2012). The details of biogenic isoprene emissions in MEGAN2.1 are described in the following section.

### 2.4 Isoprene emissions in MEGAN2.1 and two drought stress algorithms used in this study

In GEOS-Chem, biogenic isoprene emission factor ($\gamma_{2.1}$) in MEGAN2.1 is represented as below (Guenther et al., 2006; Guenther et al., 2012):

$$\gamma_{2.1} = C_{FAC}\gamma_{PAR}\gamma_T\gamma_{AGE}\gamma_{LAI}\gamma_{CO2}\gamma_{SM} = \gamma_0\gamma_{SM} \tag{1}$$

where $C_{FAC}$ is a canopy environment coefficient, $\gamma_{PAR}$ is a factor for light, $\gamma_T$ is a factor for temperature, $\gamma_{AGE}$ is a factor for leaf age, $\gamma_{LAI}$ is a factor for leaf area index (LAI), $\gamma_{CO2}$ is a factor for $CO_2$ inhibition, and $\gamma_{SM}$ is a factor for soil moisture. Here, the factors except for $\gamma_{SM}$ can be referred to as $\gamma_0$ to represent non-drought factors. $\gamma_{SM}$ is set to one by default in the standard GEOS-Chem because there is no available soil moisture database. Therefore, this limitation can lead to overestimation of isoprene

emissions and the following HCHO column under the drought conditions (Wang et al., 2022b).





We implemented two existing drought stress algorithms proposed for the better representation of isoprene emissions in MEGAN under drought conditions and tested them in the South Korean region. One drought stress algorithm implemented in this study was proposed in Wang et al. (2022b). Wang et al. (2022b) found that GEOS-Chem overestimated HCHO column in the SE US compared to the OMI HCHO column under drought conditions and this overestimation was larger in high surface

temperature conditions (above 300K). The drought stress algorithm was derived to minimize these HCHO column biases between OMI and GEOS-Chem, and the resultant formula for $\gamma_{SM}$ is shown below:

$$\gamma_{SM} = \begin{cases} 1 \ (\beta_t \geq 0.64 \ or \ T \ \leq 300K): Non-drought \\ 380.10e^{-0.02T} \ (\beta_t < 0.64 \ or \ T \ > 300K): Drought \end{cases} \tag{2}$$

where T is surface temperature and $\beta_t$ is soil moisture stress. To get $\beta_t$ value at each grid in GEOS-Chem, an ecophysiology module developed by Lam et al. (2023) was adopted in GEOS-Chem. The basic formulations in the ecophysiology module are

based on the Joint UK Land Environmental Simulator (JULES) (Best et al., 2011; Clark et al., 2011), and this ecophysiology module can calculate soil moisture stress ($\beta_t$) using gridded soil parameter data from Hadley Centre Global Environment Model version 2–Earth System Model (HadGEM2-ES). This soil moisture stress ($\beta_t$) ranges from zero (fully stressed) to one (no stress) and it can be used as a threshold to trigger the drought stress algorithm in MEGAN2.1. In this study, the threshold for the $\beta_t$ was set to 0.64 which was 60 % percentile of $\beta_t$ in the South Korean drought conditions (Fig. S2), similar to the threshold

used for the SE US (0.60) in Wang et al. (2022b).

The other drought stress algorithm implemented in this study was proposed in Jiang et al. (2018). Based on the observed physiological response of vegetations to drought stress, this drought stress algorithm was derived by reducing $V_{cmax}$, which is a maximum carboxylation rate by photosynthetic RuBisCo enzyme in vegetations (Jiang et al., 2018). The resultant formula for $\gamma_{SM}$ is shown below:

$$\gamma_{SM} = \begin{cases} 1 \ (\beta_t \geq 0.64): Non-drought \\ V_{cmax}/\alpha \ (\beta_t < 0.64, \alpha = 77): Drought \end{cases} \tag{3}$$

The $V_{cmax}$ value is also available from the ecophysiology module in GEOS-Chem. The $\alpha$ value of 77 was derived to minimize the mean bias of isoprene fluxes between GEOS-Chem and the measurement at the Missouri Ozarks AmeriFlux (MOFLUX) site under the SE US drought condition (Wang et al., 2022b).

It is noteworthy that these two drought stress algorithms were developed or tuned based on observational constraints pertaining

to the SE US region. In the next section, we investigated whether these two drought stress algorithms work in the South Korean region or not. Given that the HCHO column is the main constraint of isoprene emissions, the investigation focused on the comparison between the OMI HCHO column and the HCHO column simulated with these two drought stress algorithms. For





convenience purposes, the simulations with drought stress algorithms by Wang et al. (2022b) and by Jiang et al. (2018) were referred to as WD and JD, respectively.

## 3. HCHO column biases under the drought conditions in South Korea

### 3.1 HCHO column biases in the standard GEOS-Chem

Before investigating the two drought stress algorithms, we first compared the HCHO columns from the OMI and from the standard GEOS-Chem under the normal and drought conditions in the South Korean region (Fig. 2). The mean OMI HCHO column in South Korea was $1.11 \times 10^{16}$ molec. cm$^{-2}$ under the normal condition during the study period (2016 – 2018 JJA) and increased by 5.4 % to $1.17 \times 10^{16}$ molec. cm$^{-2}$ under the drought condition (Figs. 2a-c and Table 1). Wasti and Wang (2022) showed comparable increases of the OMI HCHO column under the South Korean drought conditions, which were 2.97 % in the mild drought and 8.02 % in the extreme drought. Since our study period was only three summers compared to that of 14 summers in Wasti and Wang (2022), we did not separate between drought severity. An increase of similar percentages in HCHO columns was also found in other regions under drought conditions such as the SE US and Amazon (Li et al., 2022; Morfopoulos et al., 2022; Wang et al., 2022b). By comparison, the mean HCHO columns simulated by the standard GEOS-Chem was $1.33 \times 10^{16}$ molec. cm$^{-2}$ under the normal condition and increased by 20.23 % to $1.59 \times 10^{16}$ molec. cm$^{-2}$ under the drought condition (Figs. 2d-f and Table 1). Thus, the model was found to significantly overestimate the increase of the HCHO column under the drought condition compared to that of the OMI HCHO column. This overestimation can be further revealed by the model biases under both normal and drought conditions. The model (relative) biases of the HCHO column were $0.22 \times 10^{16}$ molec. cm$^{-2}$ (19.82 %) under the normal condition and $0.42 \times 10^{16}$ molec. cm$^{-2}$ (35.89 %) under the drought condition. The model bias increased by $0.20 \times 10^{16}$ molec. cm$^{-2}$ (16.07 %) under the drought condition, indicating larger HCHO column bias under the drought condition. The worsening performance of GEOS-Chem under the drought condition was also found in other regions. In the SE US region (Wang et al., 2022b), for example, GEOS-Chem had a minimal bias of the HCHO column ($0.05 \times 10^{16}$ molec. cm$^{-2}$) under the normal condition, and the biases tended to increase with the drought severity ($0.08 \times 10^{16}$ molec. cm$^{-2}$ – $0.15 \times 10^{16}$ molec. cm$^{-2}$), resulting in 1.1 – 1.5 times higher increases of the HCHO column under the drought conditions compared to that of the OMI HCHO column.



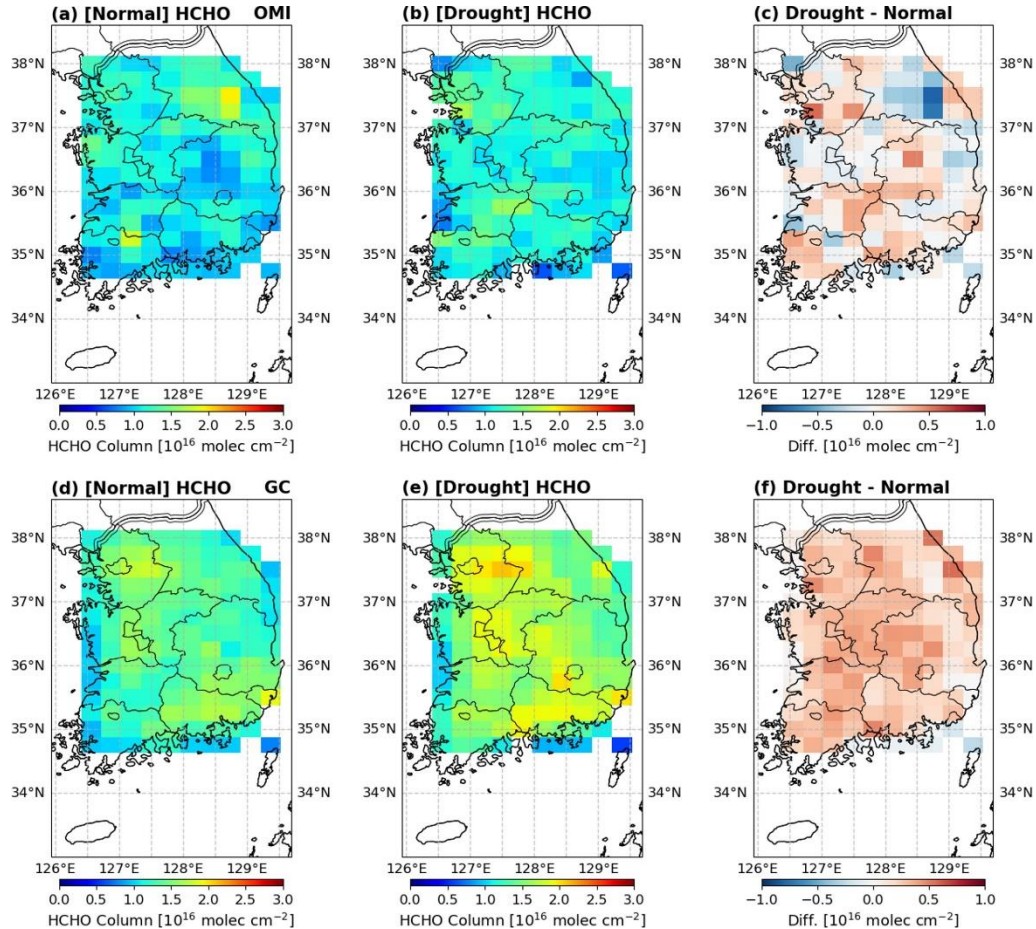

Figure 2: The OMI HCHO column under (a) the normal condition, (b) the drought condition, and (c) the difference (drought – normal). (d-f) Same as a-c but for the standard GEOS-Chem.

190

Table 1: The isoprene emissions and HCHO column under the normal condition and drought condition in South Korean region.

| | OMI | Standard GEOS-Chem | WD | JD |
|---|---|---|---|---|
| Mean isoprene emissions [$10^{-10}$kg m$^{-2}$s$^{-1}$] | | | | |
| Normal (N0) | - | 3.76 | 3.61 | 3.11 |
| Drought | - | 6.45 | 6.02 | 4.91 |



| | | (+71.5 %) | (+66.75 %) | (+57.88 %) |
|---|---|---|---|---|
| Total amount of isoprene emissions [Gg/week] | | | | |
| Normal (N0) | - | 25.08 | 24.13 | 20.72 |
| Drought | | 42.47 | 39.63 | 32.27 |
| (Compared to N0) | - | (+68.54 %) | (+64.24 %) | (+55.74 %) |
| Mean HCHO column [$10^{16}$ molec. cm$^{-2}$] | | | | |
| Normal (N0) | 1.11 | 1.33 | 1.29 | 1.24 |
| Drought | 1.17 | 1.59 | 1.53 | 1.43 |
| (Compared to N0) | (5.4 %) | (+20.23 %) | (+18.99 %) | (+15.28 %) |

As described in the aforementioned study (Wang et al., 2022b), the worsening performance of GEOS-Chem under the drought condition suggested an improper representation of the drought-induced change in the model. Given that HCHO is produced by oxidation of VOCs emitted to the atmosphere, we examined how the anthropogenic VOC (AVOC) and BVOC emissions over South Korea changed under the drought condition compared to the normal condition in the model (Fig. 3a). Here, following Lee and Park (2022), AVOC emissions included toluene, xylene, lumped alkanes (≥ C4), benzene, acetaldehyde, lumped alkene (≥ C3), $C_2H_6$, $C_3H_8$, and HCHO. BVOC emissions included isoprene, acetone, acetaldehyde, and lumped alkene (≥ C3). In the model, the total amount of AVOC emissions was consistent throughout the normal and drought conditions while that of BVOC emissions increased by 65.55 % under the drought condition compared to the normal condition. Therefore, AVOC emission changes could be ruled out and it is the increase of BVOC emissions that caused the large increase of HCHO columns in GEOS-Chem under drought conditions. Since the model has a large overestimation of HCHO columns under drought, BVOC emissions were likely overestimated under drought conditions. Wildfires are another source that can cause HCHO column bias in the model (Alvarado et al., 2020; Liao et al., 2021). However, most of the wildfires in South Korea occur in the winter-spring season and there was no major wildfire in South Korea in the study period (2016 – 2018 JJA) (Jo et al., 2023).





Given that isoprene emissions were the most dominant BVOC emissions (more than 85 %) in South Korea (Fig. S3), we examined isoprene emissions in the standard GEOS-Chem model by comparing their spatial distributions under the normal and drought conditions (Figs. 3b-d). The mean (total) fluxes of isoprene emissions over South Korea were $3.76\times10^{-10}$ kg m$^{-2}$ s$^{-1}$ (25.08 Gg/week) under the normal condition and $6.45\times10^{-10}$ kg m$^{-2}$ s$^{-1}$ (42.47 Gg/week) under the drought condition (Table 1). Notably, the mean (total) fluxes of isoprene emissions under the drought condition were 71.5 % (68.54 %) higher compared to the normal condition. This increase of isoprene emissions was larger than that of the simulated HCHO column (Table 1). The difference could be caused by enhanced HCHO loss by the increased photolysis under the clear sky during the drought condition (Naimark et al., 2021; Wang et al., 2022b). Spatially, the increase of isoprene emissions is the largest over the eastern and central parts of the region (Fig. 3d). These areas correspond to two mountain ranges in South Korea, the TaeBaek (TB) and the SoBaek (SB) mountain ranges, respectively (Yu et al., 2023). The high isoprene emissions in these regions are due to the high densities of broadleaf deciduous temperate trees, which are well-known sources of biogenic isoprene (Jang et al., 2020; Yu et al., 2023).

Despite the significant increase of isoprene emissions in the mountain ranges under the drought condition, the increase of the HCHO column under the drought condition was relatively homogeneous in the model (Fig. 2f). The HCHO production depends on the abundance of oxidants such as nitrogen oxides ($NO_x = NO_2 + NO$). In the low $NO_x$ environment, high isoprene emissions can suppress hydroxyl radicals (OH), leading to lower HCHO yield than in the high $NO_x$ environment (Marais et al., 2012; Wells et al., 2020). Indeed, we computed a ratio between the modeled HCHO column change (Drought – Normal) and the modeled isoprene emissions change (Fig. S4a) and found that the ratio tended to be higher over the regions with higher $NO_2$ column (northwest and southeast) compared to the mountain ranges with lower $NO_2$ column (Figs. S4b-c). Therefore, the different HCHO productions under the different abundance of $NO_x$ could result in the relatively homogeneous distribution of the modeled HCHO column compared to that of isoprene emissions. Given that the modeled HCHO column was affected by the abundance of $NO_x$, the model performance on $NO_2$ could play a role on the larger HCHO column bias under the drought condition (Table 1). However, GEOS-Chem was found to have only small $NO_2$ column bias (less than -3.0%) in the South Korean region and the bias was smaller under the drought condition than the normal condition (Fig. S4d). Therefore, larger HCHO column bias under the drought condition was unlikely caused by $NO_2$ bias.

Based on the above discussions, the larger HCHO column bias under the drought condition was likely caused by the model overestimation of isoprene emissions under the drought condition in South Korea (71.5 % increase; Table 1). As described in Section 2.4, MEGAN2.1 implementation in the standard GEOS-Chem model does not have the soil moisture parameter and hence overestimates isoprene emissions under the drought condition. The drought stress algorithms (i.e., WD and JD; Section 2.4) were developed to address this caveat. For example, Wang et al. (2022b) showed that the increases of isoprene emissions under the SE US drought conditions were 22.7 – 56 % in GEOS-Chem and those were reduced to 15.2 – 35.23 % after




implementing their drought stress algorithm. Given this, we implemented two existing drought stress algorithms (WD and JD; Section 2.4) in the following section and investigated whether they could reduce the overestimation of isoprene emissions and

correct the HCHO column bias under the drought condition in South Korea.

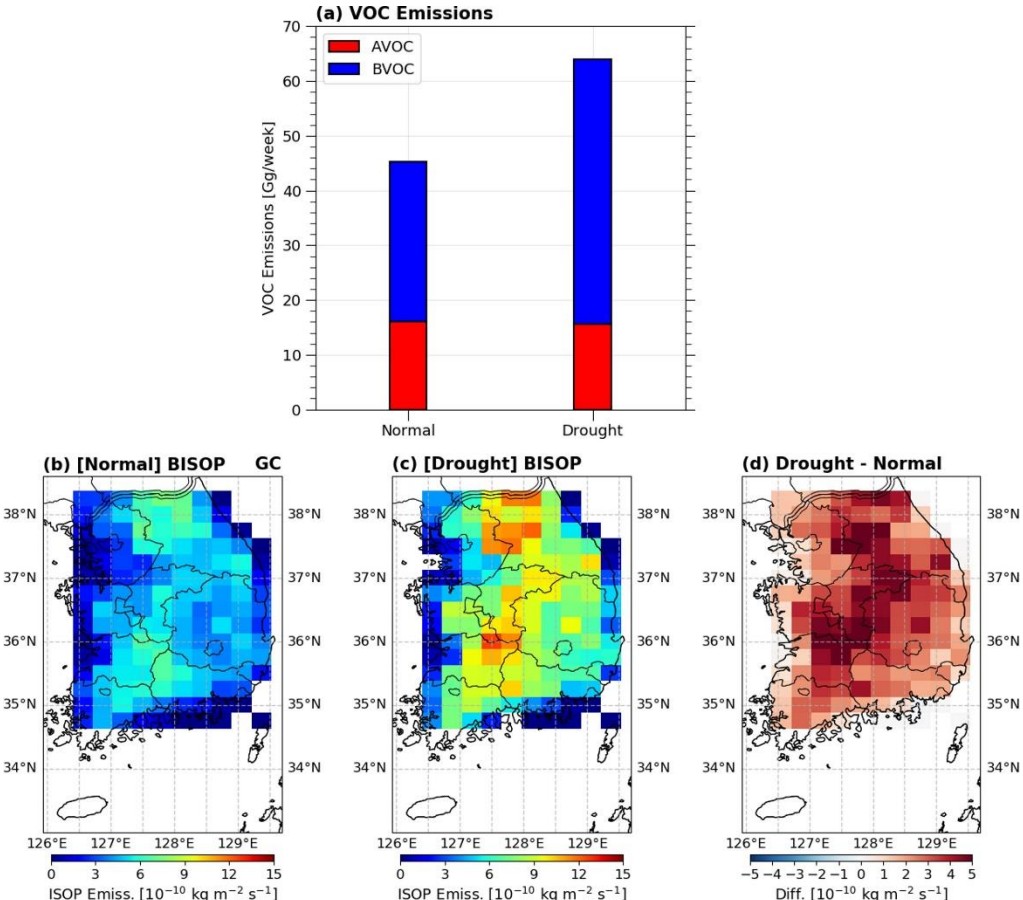

**Figure 3: (a) The total amounts of AVOC and BVOC emissions over South Korea under the normal and drought conditions. (b) The biogenic isoprene emissions (BISOP) under (b) the normal condition, (c) the drought condition, and (d) the difference (drought – normal) in the standard GEOS-Chem.**


### 3.2 HCHO column biases in two existing drought stress algorithms

Figures 4a-f show the spatial distributions of isoprene emissions simulated by WD and JD. Both WD and JD showed significant increases of isoprene emissions over the mountain ranges under the drought condition, which was consistent with the standard GEOS-Chem, but the magnitudes of the increases were reduced. In WD, the mean (total) fluxes of isoprene emissions were





$3.61\times10^{-10}$ kg m$^{-2}$s$^{-1}$ (24.13 Gg/week) under the normal condition and $6.02\times10^{-10}$ kg m$^{-2}$s$^{-1}$ (39.63 Gg/week) under the drought condition (Figs. 4a-c and Table 1). Compared to the standard GEOS-Chem, under the normal condition, the mean flux of isoprene emissions was reduced by $0.15\times10^{-10}$ kg m$^{-2}$s$^{-1}$ (3.9 %) and the total isoprene emission was reduced by 0.95 Gg/week (3.78 %). Under the drought condition, the mean flux of isoprene emissions was reduced by $0.43\times10^{-10}$ kg m$^{-2}$s$^{-1}$ (6.67 %) and the total isoprene emission was reduced by 2.84 Gg/week (6.69 %). The increase of the mean (total) fluxes of isoprene

emissions under the drought condition was 66.75 % (64.24 %), which was reduced by 4.3 % compared to the standard GEOS-Chem (Table 1). In JD, the mean (total) fluxes of isoprene emissions were $3.11\times10^{-10}$ kg m$^{-2}$s$^{-1}$ (20.72 Gg/week) under the normal condition and $4.91\times10^{-10}$ kg m$^{-2}$s$^{-1}$ (32.27 Gg/week) under the drought condition (Figs. 4d-f and Table 1). Compared to the standard GEOS-Chem, under the normal condition, the mean flux of isoprene emissions was reduced by $0.65\times10^{-10}$ kg m$^{-2}$s$^{-1}$ (17.29 %) and the total isoprene emission was reduced by 4.36 Gg/week (17.38 %). Under the drought condition, the mean

flux of isoprene emissions was reduced by $1.54\times10^{-10}$ kg m$^{-2}$s$^{-1}$ (23.88 %) and the total isoprene emission was reduced by 10.20 Gg/week (24.02 %). The increase of the mean (total) fluxes of isoprene emissions under the drought condition was 57.88 % (55.74 %), which was reduced by 12.8 % compared to the standard GEOS-Chem (Table 1).

The spatial distributions of the HCHO columns in WD and JD are presented in Figures 4g-l. In WD, the mean HCHO columns (model bias) in South Korea were $1.29\times10^{16}$ molec. cm$^{-2}$ ($0.18\times10^{16}$ molec. cm$^{-2}$; 16.22 %) under the normal condition and

$1.53\times10^{16}$ molec. cm$^{-2}$ ($0.36\times10^{16}$ molec. cm$^{-2}$; 30.77 %) under the drought condition (Figs. 4g-i and Table 1). In JD, they were $1.24\times10^{16}$ molec. cm$^{-2}$ ($0.13\times10^{16}$ molec. cm$^{-2}$; 11.71 %) under the normal condition and $1.43\times10^{16}$ molec. cm$^{-2}$ ($0.26\times10^{16}$ molec. cm$^{-2}$; 22.22 %) under the drought condition (Figs. 4j-l and Table 1). Although WD and JD reduced the biases of the HCHO column in the standard GEOS-Chem under both normal and drought conditions, the biases were still significant in those simulations especially under the drought condition (Table 1). Furthermore, both drought stress algorithms simulated a

much larger increase of the HCHO columns between normal and drought conditions (18.99 % in WD and 15.28 % in JD) than that indicated by OMI (5.4 %). Therefore, neither WD nor JD was effective for improving isoprene emissions under drought conditions in South Korea. This is because both algorithms were developed and tuned based on the SE US region as described in Section 2.4. For example, the formula for WD was derived to minimize the HCHO column biases in the SE US where the biases of the HCHO column under drought conditions were effectively reduced from $0.08\times10^{16} - 0.15\times10^{16}$ molec. cm$^{-2}$ to -

$0.05\times10^{16} - 0.02\times10^{16}$ molec. cm$^{-2}$ by implementing WD (Wang et al., 2022b). In South Korea, however, the bias of the HCHO column under the drought condition was reduced from $0.42\times10^{16}$ molec. cm$^{-2}$ only to $0.36\times10^{16}$ molec. cm$^{-2}$ by WD, indicating that the bias of the HCHO column was not effectively reduced by WD.




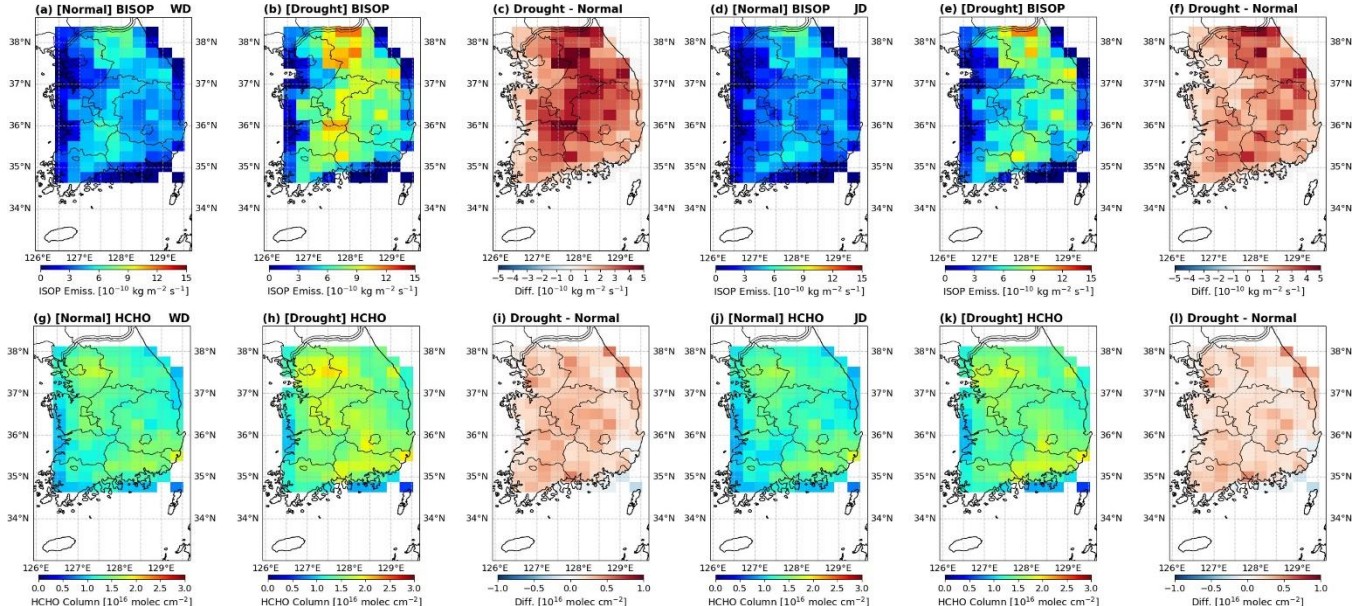

**Figure 4: The biogenic isoprene emissions (BISOP) in WD under (a) the normal condition, (b) the drought condition, and (c) the difference (drought – normal). (d-f) Same as a-c but in JD. The HCHO column in WD under (g) the normal condition, (h) the drought condition, and (i) the difference (drought – normal). (j-l) Same as g-i but in JD.**

## 4. Estimation of isoprene emissions by applying an iterative finite difference mass balance (IFDMB) method

### 4.1 Iterative finite difference mass balance (IFDMB) method

Since the overestimation of isoprene emissions under the drought condition still existed with these two drought stress algorithms, alternative approaches are needed to reduce this overestimation. An inverse modeling method using the satellite HCHO column as top-down constraint could be a useful tool to estimate isoprene emissions. In this section, we applied one of the most widely used inverse modeling methods, an iterative finite difference mass balance (IFDMB) method, to quantify isoprene emissions more accurately and to improve the HCHO column biases in the South Korean region.

The IFDMB method is one of the widely used inverse modeling methods to estimate the targeted emissions at the surface from the satellite observation of the associated atmospheric chemical species (Choi et al., 2022; Cooper et al., 2017; Li et al., 2019; Momeni et al., 2024). This study focuses on estimating isoprene emissions by utilizing the HCHO column data derived from OMI satellite observations and GEOS-Chem model simulations. The IFDMB method calculates a scaling factor by assessing the difference between the observed and modeled HCHO columns, incorporating the sensitivity of the modeled HCHO columns to isoprene emissions in each iteration. This scaling factor is subsequently applied to refine isoprene emissions. Based on its simplicity, the IFDMB method is less computationally demanding and can be more efficient (Li et al., 2019) compared





to other inverse modeling methods such as a four-dimensional variational data assimilation (4D-Var) method or a hybrid IFDMB-4DVar (Choi et al., 2022; Cooper et al., 2017). A limitation of the IFDMB method is that it neglects the effect of horizontal transport across the model columns, making it most suitable for chemical species with relatively short lifetimes. The lifetime of the isoprene and the HCHO in the atmosphere is known to be short (only a few hours) (Bates and Jacob, 2019),

so we used the IFDMB method in this study taking advantage of its simplicity. The detailed estimation of the isoprene emission at each grid and iteration in the IFDMB method is shown below (Cooper et al., 2017; Li et al., 2019; Momeni et al., 2024):

$$E_t = E_a \left( 1 + \frac{1}{S} \frac{\Omega_O - \Omega_a}{\Omega_a} \right) \tag{4}$$

where $E_t$ is the newly estimated isoprene emissions at the present iteration (a posterior), $E_a$ is the isoprene emissions at the previous iteration (a priori), $\Omega_O$ is the OMI HCHO column, $\Omega_a$ is the HCHO column simulated by GEOS-Chem, and $S$ is the

sensitivity of the HCHO column with respect to isoprene emission change. $S$ is defined at each grid and iteration as follows:

$$S = \frac{\Delta\Omega/\Omega}{\Delta E/E} \tag{5}$$

where $E$ is the biogenic isoprene emission, $\Delta E$ is a change in the biogenic isoprene emission, $\Omega$ is the HCHO column, and $\Delta\Omega$ is a change in the HCHO column. To get the initial value of $S$, an 20 % increase was applied to isoprene emissions simulated by MEGAN2.1 at each grid (Momeni et al., 2024). The $S$ was obtained in every month in all 3 years. To avoid negative values

in the posterior emissions, the constraint $(\frac{1}{S} \frac{\Omega_O - \Omega_a}{\Omega_a} > -1)$ was set so that $E_t$ cannot be negative. In addition, to make consistency with other isoprene emission dataset, the maximum value of $E_t$ at each grid was set to long-term (2005 – 2014) daily maximum isoprene emissions from the GlobEmission data (Bauwens et al., 2016) which is another isoprene emission dataset estimated from the OMI HCHO column retrievals. The iteration continued until the normalized mean difference between $E_a$ and $E_t$ was less than 1 % (Momeni et al., 2024). It required 5–7 iterations for each month to get the final posterior isoprene emissions. At

every iteration, GEOS-Chem was set to read the newly updated isoprene emissions. Other biogenic emissions such as Limonene, Ethanol, Acetaldehyde, Acetone, Sesquiterpene, Alkenes, Monoterpenes, and MTPA, and all AVOC emissions were set to the same values as in the standard GEOS-Chem simulation to focus on isoprene emission changes.

**4.2 Estimated isoprene emissions by the IFDMB and improvement in the HCHO column biases**

Figures 5a-c show the distributions of the final posterior isoprene emissions estimated by the IFDMB. The overall distribution

under the normal condition (Fig. 5a) was characterized by the high isoprene emissions over the TB and SB mountain ranges with high densities of biogenic isoprene sources, which was consistent with that from the standard GEOS-Chem (Fig. 3b). The increase of the isoprene emissions under the drought condition was also dominant over these mountain ranges (Figs. 5b-c). Thus, the spatial distribution of isoprene emissions estimated by the IFDMB was consistent with the geographical





characteristics over the South Korean region. However, as expected by the positive HCHO column biases in the standard
GEOS-Chem (Fig. 2), the IFDMB estimated the isoprene emissions over most of the South Korean region to be lower (Figs.
5a-b) compared to those in the standard GEOS-Chem (Figs. 3b-c) under both normal and drought conditions. The mean (total)
fluxes of isoprene emissions over South Korea were $1.87 \times 10^{-10}$ kgm$^{-2}$ s$^{-1}$ (12.50 Gg/week) under the normal condition and
$2.59 \times 10^{-10}$ kgm$^{-2}$ s$^{-1}$ (17.03 Gg/week) under the drought condition. Compared to the standard GEOS-Chem (Table 1), isoprene
emissions were reduced by 50.16 % under the normal condition, and by 60 % under the drought condition. The increase of
isoprene emissions under the drought condition derived by the IFDMB was 36.24 %, much lower than that in the standard
GEOS-Chem (68.54 %) and the two drought stress algorithms (64.24 % in WD, 55.74 % in JD) (Table 1). This amount of the
increase of isoprene emissions during drought was consistent with those in other regions such as a 30 % increase in northern
China (Wang et al., 2021) and 15 – 35 % increases in SE US region (Wang et al., 2022b).

The spatial distributions of the HCHO column derived from the final posterior isoprene emissions are presented in Figs. 5d-f.
The mean HCHO columns in South Korea were $1.12 \times 10^{16}$ molec. cm$^{-2}$ under the normal condition and $1.24 \times 10^{16}$ molec. cm$^{-2}$ under the drought condition. The increase of the HCHO column under the drought condition was 10.71 %, comparable to
that in OMI and much lower than that in the standard GEOS-Chem (20.23%; Table 1). Figure 5g compares the HCHO columns
under the normal and drought conditions in all simulations with OMI. As expected, the smallest HCHO column bias was found
in the IFDMB under both normal and drought conditions. The HCHO column biases of the IFDMB were $0.01 \times 10^{16}$ molec.
cm$^{-2}$ under the normal condition and $0.07 \times 10^{16}$ molec. cm$^{-2}$ under the drought condition. Under the normal condition, the mean
HCHO column bias with respect to OMI was 0.9 % in the IFDMB but the biases were 19.82 %, 16.22 %, and 11.71 % in the
standard GEOS-Chem, WD, and JD, respectively. Under the drought condition, the mean HCHO column bias with respect to
OMI was 5.98 % in the IFDMB compared to 35.9 %, 30.77 %, and 22.22 % in the standard GEOS-Chem, WD, and JD,
respectively. Therefore, by adopting the IFDMB-derived biogenic isoprene emissions, the HCHO column biases were reduced
under both normal and drought conditions in the South Korean region. This indicates that the IFDMB is effective to reduce
the HCHO column biases by adjusting isoprene emissions than the two existing drought stress algorithms.



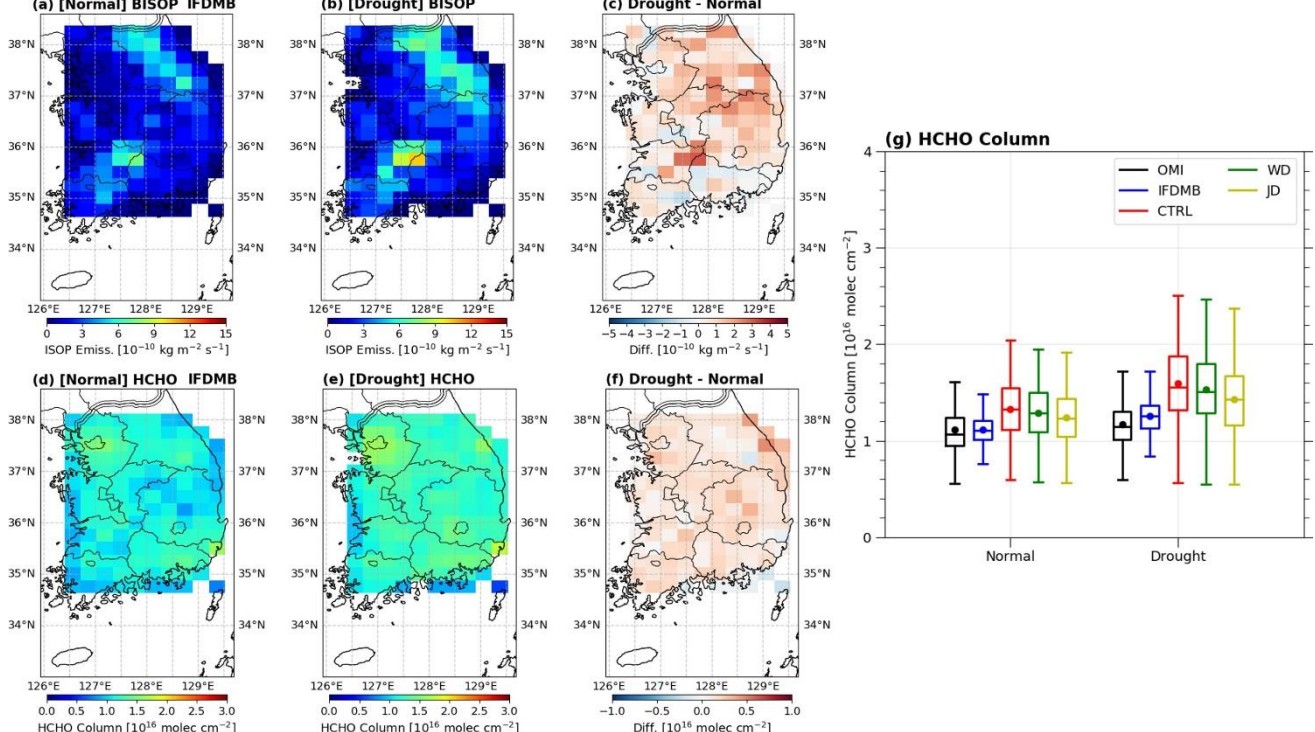

**Figure 5: The final posterior biogenic isoprene emissions (BISOP) under (a) the normal condition, (b) the drought condition, and (c)**
**the difference (drought – normal) estimated by the IFDMB. The HCHO column under (d) the normal condition, (e) the drought**
**condition, and (f) the difference (drought – normal) derived from the final posterior BISOP estimated by the IFDMB. (g) The boxplot**
**for HCHO column in OMI and in all GEOS-Chem simulations.**

## 4.3 Other secondary aerosols such as O₃ and PM₂.₅ in IFDMB

The changes in isoprene emissions can also lead to changes in other secondary air pollutants such as $O_3$ and $PM_{2.5}$. Regarding
$O_3$, previous studies showed that most of the South Korean region is in $NO_x$ saturated (VOC-limited) regime or transitional
regime for $O_3$ formation (Kashfi Yeganeh et al., 2024; Schroeder et al., 2020; Wasti and Wang, 2022). In these regimes, the
decrease (increase) in HCHO can lead to decrease (increase) in $O_3$ concentration. Given that there was no significant difference
in $NO_2$ column between the standard GEOS-Chem and the IFDMB (Fig. S4d), we investigated how the modeled $O_3$
concentrations changed with posterior isoprene emissions estimated by the IFDMB. Figures 6a-b show daytime (7am – 6pm)
mean $O_3$ concentration under the normal condition and the drought condition in the standard GEOS-Chem and its biases with
respect to the surface $O_3$ measurements. The mean observed $O_3$ concentrations in South Korea were 33.56 ppbv under the
normal condition and 42.33 ppbv under the drought condition. The increase in $O_3$ concentrations under the drought condition
was consistent with the expectation of a VOC-limited regime in response to increasing HCHO yet no change in $NO_2$ under the



drought condition as seen by OMI (Table 1 and Fig. S4d). The mean $O_3$ concentrations in the standard GEOS-Chem were 43.41 ppbv under the normal condition and 51.58 ppbv under the drought condition. The standard GEOS-Chem had positive $O_3$ biases in most of the measurement sites. The normalized mean biases were 32.24 % under the normal condition and 25.36 % under the drought condition. Figures 6c-d show the modeled $O_3$ concentrations decreased in most of the South Korean region after using posterior isoprene emissions estimated by the IFDMB. The mean $O_3$ concentrations in the IFDMB were 40.93 ppbv

under the normal condition and 46.44 ppbv under the drought condition. By applying IFDMB, therefore, the mean $O_3$ concentrations were reduced by 2.48 ppbv (5.71 %) under the normal condition and by 5.14 ppbv (9.96 %) under the drought conditions with respect to the standard GEOS-Chem. This is consistent with the VOC-limited regime where the reduction in isoprene emissions by the IFDMB leads to reduced ozone concentrations. The normalized mean biases of surface ozone in the IFDMB were 24.64 % under the normal condition and 12.98 % under the drought condition, reduced by 7.6 % and 12.38 %

respectively compared to the standard GEOS-Chem. Thus, the modeled $O_3$ concentration was found to be improved by applying the IFDMB method for better isoprene emissions modeling.

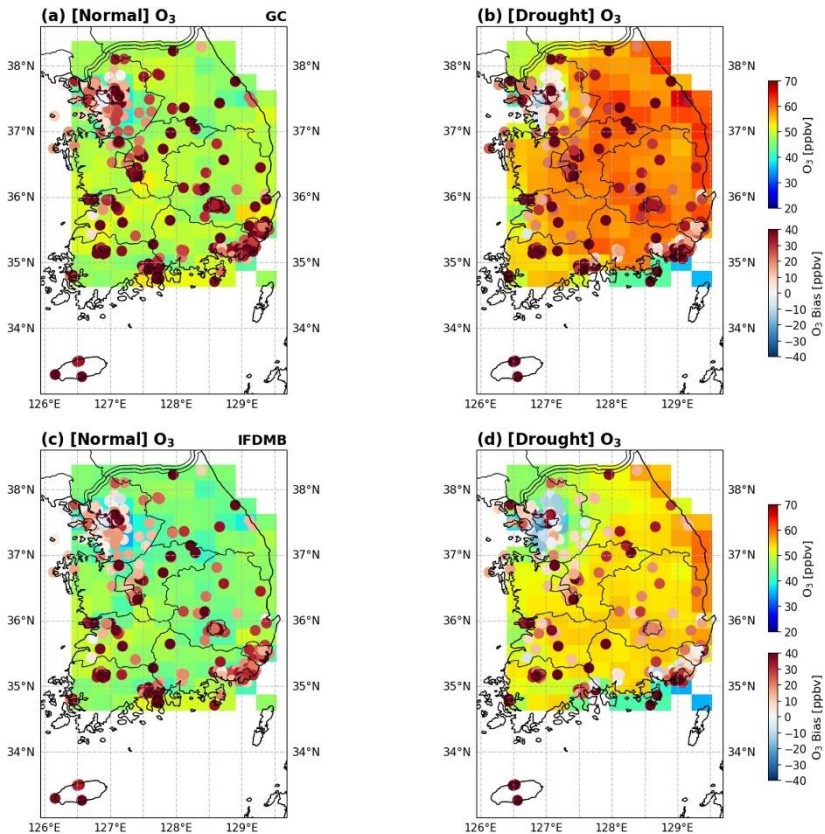

**Figure 6: The simulated $O_3$ concentrations under (a) the normal and (b) drought conditions in the standard GEOS-Chem (Shading) and the bias (GEOS-Chem–measurement; dot) at the $O_3$ measurement site. (c-d) Same as a-b but for IFDMB.**




Figures 7a-b show daytime (7am – 6pm) PM$_{2.5}$ concentrations under the normal condition and the drought condition in the standard GEOS-Chem and their biases compared to surface PM$_{2.5}$ measurements. The mean observed PM$_{2.5}$ concentrations were 17.32 μg/m$^3$ under the normal condition and 21.92 μg/m$^3$ under the drought condition. The standard GEOS-Chem had negative PM$_{2.5}$ biases in most of the measurement sites except for the Seoul metropolitan area. The mean PM$_{2.5}$ concentrations in the standard GEOS-Chem were 14.02 μg/m$^3$ under the normal condition and 15.85 μg/m$^3$ under the drought condition. The

normalized mean biases were -16.10 % under the normal condition and -25.58 % under the drought condition. Figures 7c-d are the corresponding figure for the IFDMB. Contrary to ozone, the changes in PM$_{2.5}$ concentration were not significant, which was less than 0.5 μg/m$^3$, despite the changes in isoprene emissions by the IFDMB. In South Korea, the major constituents in PM$_{2.5}$ are the inorganic constituents such as nitrate (NO$_3^-$), sulfate (SO$_4^{2-}$), ammonium (NH$_4^+$) emitted from the anthropogenic sources (Ryou et al., 2018; Park et al., 2020), which was the reason for the insignificant change in PM$_{2.5}$ between the standard

GEOS-Chem and the IFDMB.

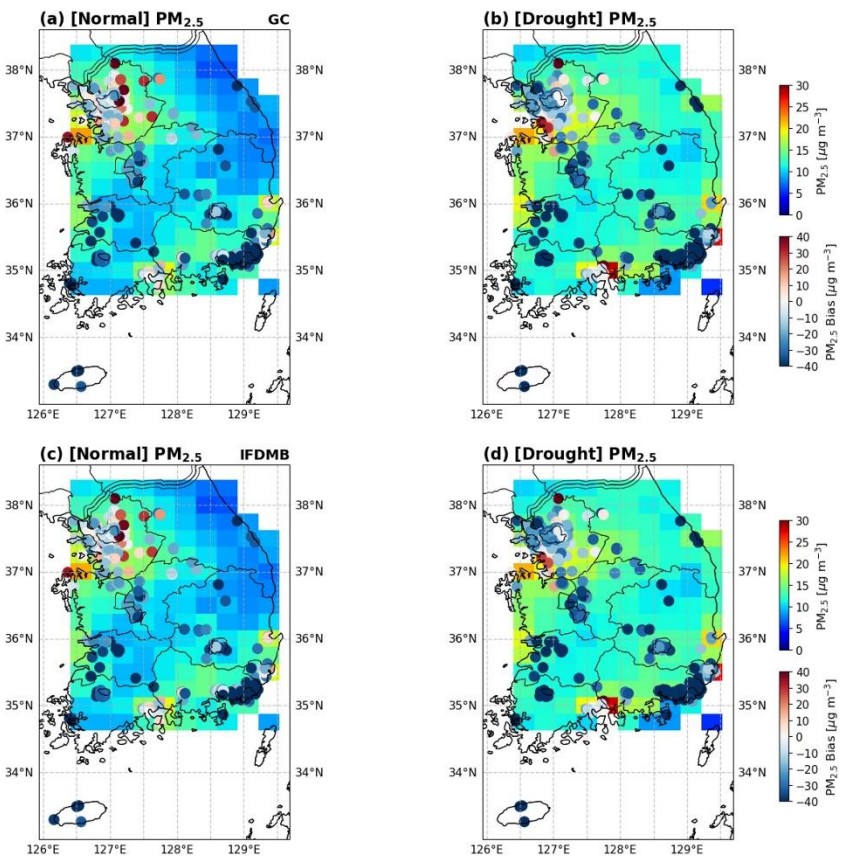

**Figure 7. Same as Fig. 6 but for PM$_{2.5}$.**





## 4.4 Empirical equations to adjust MEGAN2.1 isoprene emissions in South Korea

Based on the IFDMB results, we derived empirical equations to adjust summertime (JJA) MEGAN2.1 isoprene emissions in
South Korea in order to generalize a quantitative estimate of the drought impact on biogenic isoprene emissions in the region.
Figure 8a shows the scatterplots between biogenic isoprene emissions scaled by LAI in the standard GEOS-Chem and in
IFDMB under both normal and drought conditions. The scaling by the LAI was to consider the regional difference in biogenic
isoprene emissions due to the different vegetation coverage (Wang et al., 2022b). We overlaid the surface temperatures at each

point because it is known that isoprene emissions generally increase with the surface temperature in MEGAN2.1 (Guenther et
al., 2012; Guenther et al., 2006). The significant differences between isoprene emissions in the standard GEOS-Chem and in
the IFDMB were found mostly in conditions with high surface temperatures (Figs. 8a-b). This suggests that MEGAN2.1
implemented in the standard GEOS-Chem tends to overestimate isoprene emissions compared to those in the IFDMB in high
surface temperatures under both normal and drought conditions.

Keeping this information, we constructed an equation to adjust MEGAN2.1 isoprene emissions. The equation was based on
the difference between isoprene emissions from MEGAN2.1 and IFDMB, and was defined as follow:

$$E_{MEGAN}\left(1 + \frac{E_{IFDMB} - E_{MEGAN}}{E_{MEGAN}}\right) = E_{MEGAN}\gamma_{SM\_OMI} \tag{6}$$

where $E_{MEGAN}$ is the isoprene emissions in the standard GEOS-Chem, $E_{IFDMB}$ is the isoprene emissions estimated by the IFDMB,
and $\gamma_{SM\_OMI}$ is for the term in the parentheses $(1 + \frac{E_{IFDMB} - E_{MEGAN}}{E_{MEGAN}})$ to adjust $E_{MEGAN}$. The term in the parentheses was referred

to as $\gamma_{SM\_OMI}$ because it is derived from OMI and plays the same role as the isoprene emission factors ($\gamma_{SM}$) in the WD and JD
algorithms (Eq. 2 and Eq. 3) to adjust MEGAN2.1 isoprene emissions. Given that the difference between $E_{IFDMB}$ and $E_{MEGAN}$
was associated with the surface temperatures (Figs. 8a-b), we plotted $\gamma_{SM\_OMI}$ with respect to the surface temperatures under
both normal and drought conditions (Figs. 8c-d). It was found that $\gamma_{SM\_OMI}$ exponentially decreased as the surface temperatures
increased, which was consistent with the fact that $E_{MEGAN}$ in higher surface temperatures needed a stronger adjustment (Figs.

8a-b). The $\gamma_{SM\_OMI}$ near 295K were close to 1 (Figs. 8c-d), which reflected similar isoprene emissions between the MEGAN2.1
and the IFDMB in lower surface temperature (Figs. 8a-b). The exponentially fitted equations for $\gamma_{SM\_OMI}$ under the normal and
drought conditions are presented in Table 2. Based on this, the empirical equations for $\gamma_{SM\_OMI}$ were obtained as below:

$$\gamma_{SM\_OMI} = \begin{cases} e^{-0.274T}e^{81.00} \ (\beta_t \geq 0.64, T > 295K): Non-drought \\ e^{-0.166T}e^{49.17} \ (\beta_t < 0.64, T > 295K): Drought \end{cases} \tag{7}$$





where T is the surface temperature, and the *e* is the exponential constant. Here, we used the same soil moisture stress threshold
($\beta_t$) to separate the normal and drought conditions. Also, we set a temperature threshold to apply these empirical equations
only for the surface temperatures above 295K because the fitted equations were obtained based on the samples above 295K.
It is noteworthy that the general form of these empirical equations for $\gamma_{SM\_OMI}$ was consistent with the equation in WD,
indicating that equations for $\gamma_{SM\_OMI}$ in this study and in WD imposed stronger adjustment to MEGAN2.1 isoprene emissions
in higher surface temperatures. However, the magnitude of the adjustment was generally stronger in this study than that in WD
given the larger HCHO column biases in WD in South Korea (Figs. 4g-i; Table 1).

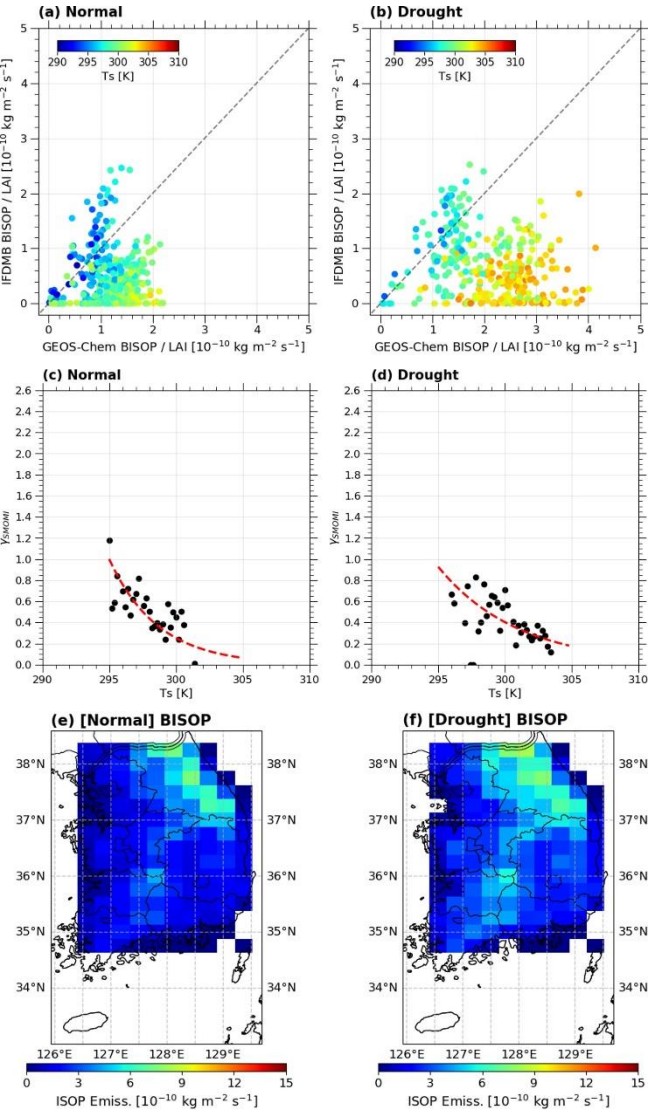



**Figure 8: The scatterplots between biogenic isoprene emissions in the standard GEOS-Chem and those in IFDMB under (a) the normal and (b) the drought conditions. Here, the biogenic isoprene emissions were divided by LAI. The surface temperature at each point was overlaid. The gray dotted line is 1:1 line. $\gamma_{SM\_OMI}$ with respect to the surface temperatures with an interval of 0.2K in (c)**

**the normal and (d) the drought conditions. The red dotted lines indicate the fitted lines. The adjusted biogenic isoprene emissions by the empirical equations for $\gamma_{SM\_OMI}$ under (e) the normal and (f) the drought conditions.**

We further verified the proposed empirical equations for $\gamma_{SM\_OMI}$ (Eq. 7) by applying them to the MEGAN2.1 isoprene emissions and getting the adjusted isoprene emissions (Figs. 8e-f). The overall structures of the adjusted isoprene emissions

were comparable to those in IFDMB under both normal and drought conditions (Figs. 5a-b). The pattern correlations between the adjusted isoprene emissions and those in the IFDMB were 0.73 under the normal condition, and 0.80 under the drought condition. The total fluxes of the adjusted isoprene emissions were 13.31 Gg/week the normal condition, 18.52 Gg/week under the drought condition, which was comparable to those in IFDMB (12.5 Gg/week under the normal condition, 17.03 Gg/week under the drought condition). Therefore, it was confirmed that applying the empirical equations for $\gamma_{SM\_OMI}$ to the MEGAN2.1

isoprene emissions could replicate isoprene emissions estimated by the IFDMB without running the inversion.

## 5. Conclusions

In this study, we showed that the mean OMI HCHO columns were $1.11 \times 10^{16}$ molec. cm$^{-2}$ under the normal condition and $1.17 \times 10^{16}$ molec. cm$^{-2}$ under the drought condition, indicating 5.4 % increase of the mean HCHO column under the drought conditions in South Korea (2016 – 2018 JJA). Compared to the OMI HCHO column, GEOS-Chem overestimated the HCHO

column by 19.82 % under the normal condition and by 35.89 % under the drought condition. The worsening performance of GEOS-Chem during drought was likely caused by the overestimation of isoprene emission increase (71.5 %) under the drought condition. Given that MEGAN2.1 implementation in the standard GEOS-Chem model does not have the soil moisture parameter and hence overestimates isoprene emissions under the drought condition, we implemented two existing drought stress algorithms (WD and JD) for the MEGNA2.1 isoprene emission. Although mean flux of isoprene emission was reduced

by 3.9 % (17.29 %) under the normal condition and by 6.67 % (23.88%) under the drought condition in WD (JD) compared to the standard GEOS-Chem, GEOS-Chem still overestimated the HCHO column by 16.22 % (11.71%) under the normal condition and by 30.77 % (22.22 %) under the drought condition in WD (JD) compared to the OMI. This result indicated that both WD and JD, which were developed and tuned based on the SE US region, were not effective to reduce HCHO column biases under the drought condition in South Korea.

To improve this, we applied an alternative approach, iterative finite difference mass balance (IFDMB) method, to estimate isoprene emissions based on the HCHO column difference between OMI and the standard GEOS-Chem. By the IFDMB method, the mean fluxes of isoprene emissions were $1.87 \times 10^{-10}$ kgm$^{-2}$ s$^{-1}$ under the normal condition and $2.59 \times 10^{-10}$ kgm$^{-2}$ s$^{-1}$



under the drought condition, which were reduced by 50.16 % under the normal condition and by 60 % under the drought condition compared to the standard GEOS-Chem. The mean HCHO columns were $1.12\times10^{16}$ molec. cm$^{-2}$ under the normal condition and $1.24\times10^{16}$ molec. cm$^{-2}$ under the drought condition. The HCHO column biases were significantly reduced to 0.9 % under normal conditions and to 5.98 % under the drought conditions with respect to the OMI. The increases of the HCHO column under the drought condition was 10.71 %, which was also comparable to the OMI.

The daytime (7am – 6pm) mean observed $O_3$ concentrations in South Korea were 33.56 ppbv under the normal condition and 42.33 ppbv under the drought condition. The standard GEOS-Chem could simulate the increase of $O_3$ under the drought condition, but it overestimated $O_3$ by 32.24 % under the normal condition and by 25.36 % under the drought condition. By applying IFDMB, these biases were improved to 24.64 % under the normal condition and to 12.98 % under the drought condition. The $PM_{2.5}$ changes between the standard GEOS-Chem and IFDMB were insignificant because major constituents in $PM_{2.5}$ in South Korea was inorganic components emitted from the anthropogenic sources.

We also proposed the empirical equations that could adjust the summertime MEGAN2.1 isoprene emissions in the South Korean region. The empirical equations were constructed based on isoprene emission difference between the standard GEOS-Chem and the IFDMB. The empirical equations were dependent on the surface temperature because significant isoprene emissions differences were found in higher surface temperatures. The adjusted isoprene emissions with these empirical equations had consistent amounts and spatial structure with those estimated by the IFDMB, which confirmed that these empirical equations could be used in the MEGAN2.1 to adjust the isoprene emissions in South Korea.

The GEOS-Chem in this study was driven by GEOS-FP meteorology dataset that is one of the conventional and widely used meteorology datasets to drive GEOS-Chem. We found that the surface temperatures from GEOS-FP meteorology have positive biases compared to the observed surface temperatures in South Korea (Fig. S5a). Given that MEGAN2.1 isoprene emissions tended to have larger biases in the higher temperature (Figs. 8a-b), these surface temperature biases might cause the overestimation of isoprene emission in the standard GEOS-Chem under both normal and drought conditions. Indeed, it was found that the regions with larger surface temperature biases tend to have larger reductions of biogenic isoprene emissions by IFDMB (Figs. S5b-c). So, isoprene emissions might be better simulated in MEGAN2.1 with the bias-corrected surface temperature in GEOS-FP meteorology dataset, which needs future investigation. In spite of this, the proposed empirical equations would be useful for the future study investigating drought stress in South Korea using GEOS-FP meteorology datasets.

As presented in this study, the air quality ($O_3$) worsens under the drought condition in South Korea. Some climate studies have showed that the drought would occur frequently with the future climate change (Zscheischler et al., 2020). So, the exquisite modeling of isoprene emissions under the drought stress is important in the air quality study. To achieve this, in-situ isoprene



emission flux measurements in the South Korea, which cover wide regions and long time periods, would be helpful to constrain the future air quality projection.

## Acknowledgements

This work was supported by NASA Atmospheric Composition Modeling and Analysis Program (80NSSC19K0986).

## Code and data availability

The GEOS-Chem version 12-7-2 is available at https://zenodo.org/records/3701669. The information for the ecophysiology module implemented in GEOS-Chem is available at https://github.com/geoschem/geos-chem/pull/629/files#diff-7df9bad6db453c401ce2a47eb0236a70f9b0951f2fcff1303e39a2cc835580c7. The DEDI is available at https://zenodo.org/records/7768534, and the observed SPI1 in South Korea is available at https://data.kma.go.kr/resources/html/en/aowdp.html. The in-situ observation of ozone and $PM_{2.5}$ in South Korea can be obtained from https://www.airkorea.or.kr/eng/. The OMI satellite HCHO and $NO_2$ columns can be obtained from https://disc.gsfc.nasa.gov/datasets/OMHCHOd_003/summary?keywords=omhchod and https://disc.gsfc.nasa.gov/datasets/OMNO2d_003/summary, respectively.

## Author contributions

YW supervised the entire research. YCJ conducted overall analysis and all model simulations using GEOS-Chem version 12-7-2 provided by HK and RJP. WL contributed to implementing the ecophysiology module and two drought stress algorithms to GEOS-Chem. MM contributed to applying the IFDMB method. YCJ wrote the manuscript draft, and all authors contributed to the preparation of the final version of the manuscript.

## Competing interests

The authors declare that they have no conflict of interest.

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
