# Peer review of "Modeling on the drought stress impact on the summertime biogenic isoprene emissions in South Korea"

_EGUsphere, 2024_

## Referee Comment (RC1)

**General Comments**

Jeong et al. investigated the impact of drought on isoprene and air quality in South Korea using OMI formaldehyde (HCHO) observations and GEOS-Chem modeling. They validated two existing drought algorithms for isoprene emissions using OMI HCHO data. Furthermore, they constrained isoprene emissions during drought using an inverse modeling approach with OMI HCHO. The topic is within the scope of ACP, and the manuscript is well organized. However, I have some concerns about the methods and techniques that need to be addressed before it can be further evaluated.

1. Bias correction of the OMI HCHO product. The author used a single correction factor of 1.28, derived from the comparison of airborne HCHO measurements from KORUS-AQ, to adjust the OMI HCHO product (Zhu et al., 2020). However, other studies using measurements from airborne platforms, FTIR, and MAX-DOAS suggest a negative bias in the OMI HCHO product when HCHO levels are high and a positive bias when HCHO levels are low (Müller et al., 2024;De Smedt et al., 2021). This phenomenon is also indicated in the reference cited by the author to justify the correction factor used (Zhu et al., 2020). I would suggest that the author consider using a different correction approach (e.g., the method in Müller et al. (2024)) for comparison considering the uncertainties from the single field campaign.

2. The impact of drought or water stress severity. The impact of water stress on isoprene emissions depends on the severity of the drought. In general, water stress can increase isoprene emissions by elevating leaf temperature through stomatal closure. However, as the drought becomes more severe, the carbon substrate supply for isoprene is cut off, leading to a decrease in emissions, as observed in field studies (Potosnak et al., 2014;Seco et al., 2015). However, the authors did not distinguish between different levels of drought severity. Therefore, I believe it is necessary to conduct an analysis based on a finer classification of drought levels.

3. The simulation of drought. Wang et al. (2022) demonstrated that the performance of drought simulations directly affects how well the model simulates the impact of drought on isoprene emissions. The authors used the soil moisture stress ($\beta t$) from the Hadley Centre Global Environment Model version 2–Earth System Model (HadGEM2-ES). However, they did not provide any information about the model's performance in capturing changes in soil moisture or water stress. Therefore, I would suggest validating the model's soil moisture or water stress simulations in the same scale of their comparison like weekly before analyzing the HCHO simulations.

4. Poor statistics. The only statistical analysis applied in the paper is the comparison of mean values, which is not sufficient for the audience to understand the analysis and the uncertainties behind these comparisons. I will provide more specific guidance on this part in my minor comments.

**Minor comments.**

Line 27: The estimation by Guenther et al., 2012 is suggesting that isoprene accounts for 50% of global BVOC emission. However, I also believe this number is quite uncertainty. So I would say 50-70% in a relatively safe way.

Line 32: Stomatal conductance" and "photosynthesis rate" are two related terms, and I don't think the statement here is correct for explaining the drought impact on isoprene emissions. In addition, Seco et al. (2022) discusses the high temperature sensitivity of isoprene in the Arctic, so I have no clue why this reference is included here.

Line 104: Please provide the referene for the OMI HCHO dataset you used.

Table 1. Please provide the standard deviation of your model results as well as the OMI HCHO column concentration, and conduct a significance test for your mean comparison.

Line 195: I assume monoterpenes and sesquiterpenes are grouped into the lumped alkenes. As indicated in Figure S3, isoprene emissions are far higher than those of other terpenoids. Could you provide more vegetation information (e.g., broadleaf and conifer tree fractions) to explain this?

Figure 6/7 and ozone/PM2.5 validation: The comparison here is quite generic and lacks details. The authors compared the model with the in-situ measurements. The only comparison presented is the Mean Bias (difference in mean values). I think some scatter plots of the model and observations on a weekly scale could be useful to understand the change in model performance after improving the model emissions using the OMI satellite data. Besides the common statistical metrics like $R^2$, RMSE, ME, and MB, a significance test should be conducted to determine if the improvement in emissions is statistically significant.

Equation 7 and Figure 8. The analysis here is confusing. I think the authors are arguing that drought stress is the main driver of the isoprene emission bias. However, the analysis focuses on temperature. Although high temperatures often coincide with drought in many cases, there are two drivers of vegetation water stress: one is the high Vapor Pressure Deficit (VPD) caused by a dry and hot atmosphere, and the other is dry soil conditions, which determine the water supply for plants. Additionally, long-lasting droughts are mainly controlled by a lack of water. However, the equation and analysis here use the soil moisture

parameter (βt) as the indicator of drought severity but use temperature as the input for addressing the isoprene emission bias. This raises the question: is the bias caused by drought, or temperature, or both?

**Reference**

De Smedt, I., Pinardi, G., Vigouroux, C., Compernolle, S., Bais, A., Benavent, N., Boersma, F., Chan, K. L., Donner, S., Eichmann, K. U., Hedelt, P., Hendrick, F., Irie, H., Kumar, V., Lambert, J. C., Langerock, B., Lerot, C., Liu, C., Loyola, D., Piters, A., Richter, A., Rivera Cárdenas, C., Romahn, F., Ryan, R. G., Sinha, V., Theys, N., Vlietinck, J., Wagner, T., Wang, T., Yu, H., and Van Roozendael, M.: Comparative assessment of TROPOMI and OMI formaldehyde observations and validation against MAX-DOAS network column measurements, Atmos. Chem. Phys., 21, 12561-12593, 10.5194/acp-21-12561-2021, 2021.

Müller, J. F., Stavrakou, T., Oomen, G. M., Opacka, B., De Smedt, I., Guenther, A., Vigouroux, C., Langerock, B., Aquino, C. A. B., Grutter, M., Hannigan, J., Hase, F., Kivi, R., Lutsch, E., Mahieu, E., Makarova, M., Metzger, J. M., Morino, I., Murata, I., Nagahama, T., Notholt, J., Ortega, I., Palm, M., Röhling, A., Stremme, W., Strong, K., Sussmann, R., Té, Y., and Fried, A.: Bias correction of OMI HCHO columns based on FTIR and aircraft measurements and impact on top-down emission estimates, Atmos. Chem. Phys., 24, 2207-2237, 10.5194/acp-24-2207-2024, 2024.

Potosnak, M. J., LeStourgeon, L., Pallardy, S. G., Hosman, K. P., Gu, L., Karl, T., Geron, C., and Guenther, A. B.: Observed and modeled ecosystem isoprene fluxes from an oak-dominated temperate forest and the influence of drought stress, Atmospheric Environment, 84, 314-322, https://doi.org/10.1016/j.atmosenv.2013.11.055, 2014.

Seco, R., Karl, T., Guenther, A., Hosman, K. P., Pallardy, S. G., Gu, L., Geron, C., Harley, P., and Kim, S.: Ecosystem-scale volatile organic compound fluxes during an extreme drought in a broadleaf temperate forest of the Missouri Ozarks (central USA), Global Change Biology, 21, 3657-3674, doi:10.1111/gcb.12980, 2015.

Seco, R., Holst, T., Davie-Martin, C. L., Simin, T., Guenther, A., Pirk, N., Rinne, J., and Rinnan, R.: Strong isoprene emission response to temperature in tundra vegetation, Proceedings of the National Academy of Sciences, 119, e2118014119, 10.1073/pnas.2118014119, 2022.

Wang, H., Lu, X., Seco, R., Stavrakou, T., Karl, T., Jiang, X., Gu, L., and Guenther, A. B.: Modeling Isoprene Emission Response to Drought and Heatwaves Within MEGAN Using Evapotranspiration Data and by Coupling With the Community Land Model, Journal of Advances in Modeling Earth Systems, 14, e2022MS003174, https://doi.org/10.1029/2022MS003174, 2022.

Zhu, L., González Abad, G., Nowlan, C. R., Chan Miller, C., Chance, K., Apel, E. C., DiGangi, J. P., Fried, A., Hanisco, T. F., Hornbrook, R. S., Hu, L., Kaiser, J., Keutsch, F. N., Permar, W., St. Clair, J. M., and Wolfe, G. M.: Validation of satellite formaldehyde (HCHO) retrievals using observations from 12 aircraft campaigns, Atmos. Chem. Phys., 20, 12329-12345, 10.5194/acp-20-12329-2020, 2020.

---

## Referee Comment (RC2)

**Modeling on the drought stress impact on the summertime biogenic isoprene emissions in South Korea**

Jeong et al., 2024

This manuscript investigates the potential to constraining biogenic isoprene emissions under both normal and drought conditions by incorporating drought stress into the modelling with GEOS-chem. The study highlights that previously implemented algorithms used for the southeastern United States are not directly applicable to the South Korean peninsula. Instead, the authors utilized the IFDMB framework to derive an empirical equation tailored to the South Korean region, significantly enhancing prediction accuracy under both normal and drought conditions. Furthermore, this approach also improved the prediction of other secondary pollutants under these conditions. Throughout the study, the methods and results convincingly highlight the scientific significance of this work, offering substantial findings that underscore its value for modeling applications. Also, this research provides valuable insights to enhance model predictions in regional air quality (AQ) contexts.

I would recommend a list of points and concerns which needs to be addressed.

P1. Throughout the manuscript, the authors use absolute/percentage number in the text, making it difficult and complicated for readers to follow every detail. They should refine how these results are presented. For example, they could utilize the unused white space in the spatial plots (panels) to display relevant values or add more tables with columns showing results before and after implementing the algorithm. Overall, improving the readability of the manuscript would be a substantial value addition.

P2. It is assumed that a drought index is used solely to identify drought conditions. Why were ETDI or DEDI chosen here when other drought indices are also available? Some insight on this would be valuable because DEDI is essentially an agricultural drought index best suited for short-term developing droughts, although it can also capture longer-term drying conditions (Narasimhan and Srinivasan 2005; Singh et al., 2024). Consequently, its application to vegetation---especially forests with deeply rooted trees---may not detect short-duration events. It is assumed that the developed and tested algorithm would be applicable regardless of the drought's duration and severity. Adding some discussion on this would be helpful.

P3. This study demonstrated that the algorithm developed and proven effective over the SE US are not equally effective over the SK region. Authors can pull some insights over the possible reasons behind this. Is it due to different types of vegetation over these regions or any other factors at play? As mentioned in P2, if the emission response is moderated by different vegetations types, then improvements claimed by the new empirical algorithm may be dependent on the types (hydrological or meteorological) and severity of drought, and thus be more region-specific (e.g., algorithms over the southeastern U.S.).

P4. Authors has reported the results % change across the manuscript. Authors should also provide some estimate of uncertainties over the region. I.e sensitivity of new approach.

P5: The authors should verify the calculations for the percentage change in emissions reported in Table 1 (e.g., isoprene emissions for the standard case, and HCHO). It appears that the percentage change between normal and drought conditions was computed using values with more decimal precision than those shown in Table 1, which only displays two decimal places. Consequently, the percentage changes in Table 1 differ by about 1–2% from the values one would obtain using the tabulated data. The authors should ensure consistency and revise these figures throughout the manuscript after cross-checking.

P6. Figure 5 and its description (section 4.2) is really confusing and hard to keep track when previous figures are referred. I assume in figure 5, row 1, panel 2 (drought) should have the IFDMB? Also, Fig 5c shows the box-whiskers for HCHP different approaches for all GEOS-chem simulations. how many? What would be the significance of these bar/statistics? This needs some efforts to make it more explanatory.

P7: Line 418: "presented in table 2." Table 2 is missing in manuscript. This is vital.

Specifics: -

1. Line 10: - I suggest authors to rephrase this sentence to frame the importance of reducing the uncertainties in these emission context to SK region instead of directly stating the effort has not been made in SK region. This will better abstract the requirement and gap in knowledge in context to the region.
2. Line 40: - "some studies" sounds vague here, better cite them here. Also, in line 43 if "some previous studies" refers to the cited in the end of sentences (line 46), author can rephrase this like "recent studies…."
3. Line 112: "factor of 1.28*.. "sounds like a sudden introduction of something important. Presumably this comes from the Shen et al., 2019; Wang et al 2022b. I suggest a rephrasing of this sentence for better connectivity and explanation.
4. Line 326: "fig 3.c". please check figure 3 if "c" is marked there?
5. Line 324" "geographical characteristics over the South Korean" please elaborate the context of geographical condition here. It is related to vegetation or climatic features.

References:
Narasimhan, B. and Srinivasan, R.: Development and evaluation of Soil Moisture Deficit Index (SMDI) and Evapotranspiration Deficit Index (ETDI) for agricultural drought monitoring, Agricultural and Forest Meteorology, 133, 69–88, https://doi.org/10.1016/j.agrformet.2005.07.012, 2005.

Singh, R., Tsigaridis, K., Bull, D., Swiler, L. P., Wagman, B. M., and Marvel, K.: Mount Pinatubo's effect on the moisture-based drivers of plant productivity, EGUsphere [preprint], https://doi.org/10.5194/egusphere-2024-2280, 2024.

---

## Author Comment (AC1)

**Modeling on the drought stress impact on the summertime biogenic isoprene emissions in South Korea**

Jeong et al., 2024

This manuscript investigates the potential to constraining biogenic isoprene emissions under both normal and drought conditions by incorporating drought stress into the modelling with GEOSchem. The study highlights that previously implemented algorithms used for the southeastern United States are not directly applicable to the South Korean peninsula. Instead, the authors utilized the IFDMB framework to derive an empirical equation tailored to the South Korean region, significantly enhancing prediction accuracy under both normal and drought conditions. Furthermore, this approach also improved the prediction of other secondary pollutants under these conditions. Throughout the study, the methods and results convincingly highlight the scientific significance of this work, offering substantial findings that underscore its value for modeling applications. Also, this research provides valuable insights to enhance model predictions in regional air quality (AQ) contexts.

I would recommend a list of points and concerns which needs to be addressed.

→ We sincerely appreciate the reviewer who gave the constructive comments to improve the manuscript. Their comments are reproduced below followed by our responses in blue. The corresponding edits in the manuscript are highlighted with red color.

P1. Throughout the manuscript, the authors use absolute/percentage number in the text, making it difficult and complicated for readers to follow every detail. They should refine how these results are presented. For example, they could utilize the unused white space in the spatial plots (panels) to display relevant values or add more tables with columns showing results before and after implementing the algorithm. Overall, improving the readability of the manuscript would be a substantial value addition.

→ We agree with the reviewer's comment. In response, we have added values of the mean (total) isoprene emissions and mean HCHO column in each panel in the revised Figures 2-5. As the previous Table 1 contained these values, we changed Table 1 to the mean HCHO column bias of GEOS-Chem simulations for better readability. In addition, we added Table 2 presenting the differences in the biogenic isoprene emissions and the HCHO column simulated between each drought stress algorithms and the standard GEOS-Chem.

Table 1: The mean HCHO column bias (relative bias) of GEOS-Chem simulations under the normal condition and drought condition in South Korean region.

| Unit: $10^{16}$ molec. cm$^{-2}$ | Standard GEOS-Chem | WD | JD |
|---|---|---|---|
| Normal | 0.22 (19.82 %) | 0.18 (16.22 %) | 0.13 (11.71 %) |
| Drought | 0.42 (35.89 %) | 0.36 (30.77 %) | 0.26 (22.22 %) |

Table 2: The differences (relative difference) in the biogenic isoprene emissions and the HCHO column simulated by each drought stress algorithms compared to the standard GEOS-Chem.

| Drought stress algorithms | Drought | Mean flux of isoprene emissions [Unit: $10^{-10}$ kgm$^{-2}$s$^{-1}$] | Total isoprene emissions [Unit: Gg/week] | Mean HCHO column [Unit: $10^{16}$ molec. cm$^{-2}$] |
|---|---|---|---|---|
| WD | Normal | -0.15 (-3.9 %) | -0.95 (-3.78 %) | -0.04 (-3.00 %) |
| | Drought | -0.43 (-6.67 %) | -2.84 (-6.69 %) | -0.06 (-3.77 %) |
| JD | Normal | -0.65 (-17.29 %) | -4.36 (-17.38 %) | -0.09 (-6.77 %) |
| | Drought | -1.54 (-23.88 %) | -10.20 (-24.02 %) | -0.16 (-10.06 %) |

P2. It is assumed that a drought index is used solely to identify drought conditions. Why were ETDI or DEDI chosen here when other drought indices are also available? Some insight on this would be valuable because DEDI is essentially an agricultural drought index best suited for short-term developing droughts, although it can also capture longer-term drying conditions (Narasimhan and Srinivasan 2005; Singh et al., 2024). Consequently, its application to vegetation---especially forests with deeply rooted trees---may not detect short-duration events. It is assumed that the developed and tested algorithm would be applicable regardless of the drought's duration and severity. Adding some discussion on this would be helpful.

→ The reviewer's point is well taken. The reason why we used DEDI in this study is that it is based on the balance of evapotranspiration between atmosphere and terrestrial ecosystem (Zhang et al., 2023), which can connect the climate system and vegetations. In addition, it has a fine gridded resolution (0.25° × 0.25°) and temporal resolution (daily). This fine spatiotemporal resolution of DEDI helped us to increase a sampling size in the limited study period (2016, 2017, and 2018 JJA) and domain (South Korea). Although we used DEDI only in this study, we compared DEDI to the Standardized Precipitation Index (SPI) in the South Korea region (Figure S1) and found that DEDI is consistent with the SPI, as reported in the Zhang et al. (2023), indicating that the overall results would be consistent when SPI is used in this study. The comparison of DEDI to SPI was already discussed in the original manuscript and we added the following sentence in the revised manuscript.

"Since DEDI data is available at fine spatial (0.25° × 0.25°) and temporal (daily) resolutions than other drought indices, we chose DEDI over other indices to help increase data sizes in the limited study period and domain."

P3. This study demonstrated that the algorithm developed and proven effective over the SE US are not equally effective over the SK region. Authors can pull some insights over the possible reasons behind this. Is it due to different types of vegetation over these regions or any other factors at play? As mentioned in P2, if the emission response is moderated by different vegetations types, then improvements claimed by the new empirical algorithm may be dependent on the types (hydrological or meteorological) and severity of drought, and thus be more region specific (e.g., algorithms over the southeastern U.S.).

→ This is a good point. As the response of the isoprene emissions to the drought can be different by vegetation types (mainly deciduous-leaf trees), the empirical drought stress algorithms

based on the southeastern US may not work in South Korea. For example, in South Korea, the main deciduous-leaf trees are *Quercus mongolica*, *Quercus variabilis*, and *Quercus acutissima* (Lee et al., 2025). While in southeastern US, the main deciduous-leaf trees are *Quercus stellata, Quercus alba*, and *Quercus prinus* (Perry et al., 2022). These difference in main deciduous-leaf tree species between the southeastern US and South Korea can make drought stress algorithms based on the southeastern US not effective in South Korea. Responding to the reviewer's comment, we added the following sentences in the revised manuscript.

"The two regions have different main deciduous-leaf tree species. South Korea has mainly *Quercus mongolica*, *Quercus variabilis*, and *Quercus acutissima* (Lee et al., 2025), while the SE US has *Quercus stellata*, *Quercus alba*, and *Quercus prinus* (Perry et. al., 2022). This fundamental difference may cause ineffectiveness of WD and JD in South Korea."

P4. Authors have reported the results % change across the manuscript. Authors should also provide some estimate of uncertainties over the region. I.e sensitivity of new approach.

→ To provide statistical information, the mean, standard deviation, and p-value based on Student's *t*-test were added in each figure (Figures 2-5).

P5: The authors should verify the calculations for the percentage change in emissions reported in Table 1 (e.g., isoprene emissions for the standard case, and HCHO). It appears that the percentage change between normal and drought conditions was computed using values with more decimal precision than those shown in Table 1, which only displays two decimal places. Consequently, the percentage changes in Table 1 differ by about 1–2% from the values one would obtain using the tabulated data. The authors should ensure consistency and revise these figures throughout the manuscript after cross-checking.

→ Corrected. Revised values were added in Figures 2-5 and Tables 1-2.

P6. Figure 5 and its description (section 4.2) is really confusing and hard to keep track when previous figures are referred. I assume in figure 5, row 1, panel 2 (drought) should have the IFDMB?  Also, Fig 5c shows the box-whiskers for HCHO different approaches for all GEOSchem simulations. how many? What would be the significance of these bar/statistics? This needs some efforts to make it more explanatory.

→ We added a caption for each panel in Figure 5 for clarity in Section 4.2. The box in boxplot extends from the first quartile (Q1) to the third quartile (Q3) with a line at the median value and a dot at a mean value. The inter-quartile range (IQR) is from Q1 to Q3, and the whisker in boxplot extends from the Q1-1.5×IQR to the Q3+1.5×IQR. We added this explanation in the caption for Figure 5.

P7: Line 418: "presented in table 2." Table 2 is missing in manuscript. This is vital.

→ The original Table 2 was removed in response to a comment by Reviewer #1. There is a new Table 2 in the revised manuscript as shown above (P1).

**Specifics:**

1. Line 10: - I suggest authors to rephrase this sentence to frame the importance of reducing the uncertainties in these emission context to SK region instead of directly stating the effort has not been made in SK region. This will better abstract the requirement and gap in knowledge in context to the region.

→ Agreed. We revised the sentence as below:

"This study aims at constraining drought stress on biogenic isoprene emissions in South Korea using satellite formaldehyde (HCHO) column"

2. Line 40: - "some studies" sounds vague here, better cite them here. Also, in line 43 if "some previous studies" refers to the cited in the end of sentences (line 46), author can rephrase this like "recent studies…."

→ Yes, we revised those sentences in the revised manuscript as below:

"…, some studies (Wang et al., 2022b; Wasti and Wang, 2022) have used the tropospheric formaldehyde (HCHO) column retrievals from the satellite to estimate isoprene emissions response to drought."

"Recent studies (Li et al., 2022; Naimark et al., 2021) showed that the tropospheric HCHO column from the Ozone Monitoring Instrument (OMI) on the Aura satellite increased by 6.5–22 % in the southeastern United States (US) region during the summertime drought, which is indicative of the increase of isoprene emissions during drought."

3. Line 112: "factor of 1.28*.. "sounds like a sudden introduction of something important. Presumably this comes from the Shen et al., 2019; Wang et al 2022b. I suggest a rephrasing of this sentence for better connectivity and explanation.

→ The sentence was revised in the revised manuscript as below:

"To correct this underestimation, we applied a constant factor of 1.28 (1 / (1–0.22)) to the OMHCHOd data as in the previous studies (Shen et al., 2019; Wang et al., 2022b)."

4. Line 326: "fig 3.c". please check figure 3 if "c" is marked there?

→ Yes, Figure 3c shows the biogenic isoprene emissions under the drought condition in the standard GEOS-Chem.

5. Line 324" "geographical characteristics over the South Korean" please elaborate the context of geographical condition here. It is related to vegetation or climatic features.

→ We used the term of "geographical characteristics" here to state that the spatial distribution of isoprene emissions estimated by the IFDMB was consistent with the mountain ranges where the sources of biogenic isoprene were highly populated. Responding to the reviewer's comment, we revised the sentence as below:

"In both normal and drought conditions, therefore, the spatial distribution of isoprene emissions estimated by the IFDMB could represent higher isoprene emissions over the mountain ranges with large density of broadleaf trees."

**References:**

Narasimhan, B. and Srinivasan, R.: Development and evaluation of Soil Moisture Deficit Index (SMDI) and Evapotranspiration Deficit Index (ETDI) for agricultural drought monitoring, Agricultural and Forest Meteorology, 133, 69–88, https://doi.org/10.1016/j.agrformet.2005.07.012, 2005.

Singh, R., Tsigaridis, K., Bull, D., Swiler, L. P., Wagman, B. M., and Marvel, K.: Mount Pinatubo's effect on the moisture-based drivers of plant productivity, EGUsphere [preprint], https://doi.org/10.5194/egusphere-2024-2280, 2024.

---

## Author Comment (AC2)

**General Comments**

Jeong et al. investigated the impact of drought on isoprene and air quality in South Korea using OMI formaldehyde (HCHO) observations and GEOS-Chem modeling. They validated two existing drought algorithms for isoprene emissions using OMI HCHO data. Furthermore, they constrained isoprene emissions during drought using an inverse modeling approach with OMI HCHO. The topic is within the scope of ACP, and the manuscript is well organized. However, I have some concerns about the methods and techniques that need to be addressed before it can be further evaluated.

→ We sincerely appreciate the reviewer who gave the constructive comments to improve the manuscript. Their comments are reproduced below followed by our responses in blue. The corresponding edits in the manuscript are highlighted with red color.

1. Bias correction of the OMI HCHO product. The author used a single correction factor of 1.28, derived from the comparison of airborne HCHO measurements from KORUS-AQ, to adjust the OMI HCHO product (Zhu et al., 2020). However, other studies using measurements from airborne platforms, FTIR, and MAX-DOAS suggest a negative bias in the OMI HCHO product when HCHO levels are high and a positive bias when HCHO levels are low (Müller et al., 2024; De Smedt et al., 2021). This phenomenon is also indicated in the reference cited by the author to justify the correction factor used (Zhu et al., 2020). I would suggest that the author consider using a different correction approach (e.g., the method in Müller et al. (2024)) for comparison considering the uncertainties from the single field campaign.

→ We appreciate the reviewer's constructive comment. Responding to the reviewer's comments, we calculated the bias-corrected OMI HCHO column based on the method in Müller et al. (2024) and compared it to the originally corrected OMI HCHO column (following Zhu et al. (2020)) under both normal and drought conditions (Fig. S2 in the revised manuscript). The two corrected HCHO columns were highly correlated with a correlation slope close to 1, expect for a few low-value points deviating from the 1:1 regression line under normal conditions. This suggests a single correction factor adopted by the manuscript is acceptable. We added the following sentences and figure (Fig. S2) in the revised manuscript:

"We also compared this method to a different bias correction method suggested by Müller et al. (2024) and the results were shown in Figure S2. The two corrected HCHO columns were highly correlated with a regression slope close to one, except for a few HCHO points under normal condition. The corrected HCHO columns used in this study were 7-12% higher compared to those from Müller et al. (2024) under normal and drought conditions, respectively."

[Figure]

Figure S2: Scatterplot between two bias-corrected OMI HCHO columns following Zhu et al. (2020) and Müller et al. (2024) under (a) normal and (b) drought conditions. The method in Zhu et al. (2020) was used in this study. Each dot denotes HCHO column value at each grid point in South Korea. The gray dashed line denotes 1:1 line and the red line denotes linear regression line. The slope for the regression line is shown at the right side of the panel with the correlation coefficient (Corr.) between two HCHO columns.

2. The impact of drought or water stress severity. The impact of water stress on isoprene emissions depends on the severity of the drought. In general, water stress can increase isoprene emissions by elevating leaf temperature through stomatal closure. However, as the drought becomes more severe, the carbon substrate supply for isoprene is cut off, leading to a decrease in emissions, as observed in field studies (Potosnak et al., 2014; Seco et al., 2015). However, the authors did not distinguish between different levels of drought severity. Therefore, I believe it is necessary to conduct an analysis based on a finer classification of drought levels.

→ We agree with the review on this point. Following Zhang et al. (2022), the DEDI drought severity can be broken down into five categories (Table S1 in the revised manuscript). Based on these five categories, we calculated the observed HCHO columns in each drought category (Figure S4 in the revised manuscript). The mean HCHO columns indeed tended to increase as the drought was stronger, which is consistent with Wasti and Wang (2022) showing the 2.97 % increase of the OMI HCHO column under the mild drought and 8.02 % in the extreme drought. However, as described in the paper, since our study period included only three summers the number of cases in each drought category was small, especially for severe and extreme categories, and thus we did not elaborate on the impact of drought severity. We added the following sentences, table (Table. S1), and figure (Fig. S4) in the revised manuscript to discuss the impact of drought severity.

"It is known that the impact of water stress on isoprene emissions depends on the severity of the drought (Potosnak et al., 2014; Seco et al., 2015; Wasti and Wang, 2022). To examine this impact, the DEDI was separated into five categories following Zhang et al. (2022): Normal/Wet, Incipient drought, Moderate drought, Severe drought, and Extreme drought (Table S1). Based on these five categories, we calculated the observed HCHO columns in each drought category (Fig. S4). While the domain-mean HCHO columns tended to increase as the drought became stronger, the signal is weak and not uniform by location. For example, the OMI HCHO column over the northeastern parts of South Korea (Taebaek Mountains), which showed a decrease under the drought condition (Figs. 2a-c), showed an increase only under the extreme drought category (Fig. S4). This is probably because our study period included only three summers.

Given the small sampling size, we chose not to separate drought severity in the following analysis."

Table S1: Five drought categories by DEDI drought index

| Drought category | Chance of occurrence | DEDI threshold (South Korea) |
|---|---|---|
| Extreme drought | 2% or less | DEDI ≤ -2.02 |
| Severe drought | 2 – 10 % | -2.02 < DEDI ≤ -1.15 |
| Moderate drought | 10 – 20 % | -1.15 < DEDI ≤ -0.75 |
| Incipient drought | 20 – 30 % | -0.75 < DEDI ≤ -0.49 |
| Normal/Wet | More than 30 % | -0.49 < DEDI |

[Figure]

Figure S4: The OMI HCHO columns under five drought categories based on DEDI drought index.

3. The simulation of drought. Wang et al. (2022) demonstrated that the performance of drought simulations directly affects how well the model simulates the impact of drought on isoprene emissions. The authors used the soil moisture stress ($\beta t$) from the Hadley Centre Global Environment Model version 2–Earth System Model (HadGEM2-ES). However, they did not provide any information about the model's performance in capturing changes in soil moisture or water stress. Therefore, I would suggest validating the model's soil moisture or water stress simulations in the same scale of their comparison like weekly before analyzing the HCHO simulations.

→ Because of no available observations of soil moisture to validate the soil moisture stress ($\beta t$), we used the same approach as in Wang et al. (2022) to calibrate the soil moisture stress ($\beta t$) according to the drought index. In Figure S3 in the revised manuscript, we showed the distribution of calculated soil moisture stress ($\beta t$) under the normal and drought conditions based on DEDI drought index. Consistent with the Wang et al. (2022), we selected a threshold

of βt below which it was under drought condition. This threshold was 0.64, which was 60% percentile of βt in the South Korea drought conditions. This value was used to turn on drought stress algorithms in GEOS-Chem.

4. Poor statistics. The only statistical analysis applied in the paper is the comparison of mean values, which is not sufficient for the audience to understand the analysis and the uncertainties behind these comparisons. I will provide more specific guidance on this part in my minor comments.

→ Thank you for the reviewer's constructive comment. We responded to this comment in the reviewer's specific minor comment below.

**Minor comments.**

Line 27: The estimation by Guenther et al., 2012 is suggesting that isoprene accounts for 50% of global BVOC emission. However, I also believe this number is quite uncertainty. So I would say 50-70% in a relatively safe way.

→ Good point. We revised the manuscript as below:

"BVOCs are emitted from terrestrial vegetations and 50–70 % of global BVOCs emissions are isoprene emissions (Pacifico et al., 2009; Sindelarova et al., 2014)."

Line 32: Stomatal conductance" and "photosynthesis rate" are two related terms, and I don't think the statement here is correct for explaining the drought impact on isoprene emissions. In addition, Seco et al. (2022) discusses the high temperature sensitivity of isoprene in the Arctic, so I have no clue why this reference is included here.

→ The Seco et al. reference was used to support the statement on the overall dependence of isoprene emissions on meteorological factors. We revised the manuscript as below:

"Isoprene emissions depend on not only physiological factors such as plant functional type, leaf area index, and leaf age, but also meteorological factors such as temperature, radiation, and soil moisture, which affect plant's physiology such as stomatal conductance, isoprene synthase activity, and carbon substrate supply (Ferracci et al., 2020; Guenther et al., 2012; Guenther et al., 2006; Potosnak et al., 2014; Seco et al., 2022)."

Line 104: Please provide the reference for the OMI HCHO dataset you used.

→ Added.

Table 1. Please provide the standard deviation of your model results as well as the OMI HCHO column concentration, and conduct a significance test for your mean comparison.

→ The original Table 1 changed to the mean HCHO column bias of GEOS-Chem simulations for better readability. Instead, we added the mean, standard deviation, and p-value based on Student's $t$-test in each figure (Figures 2-5).

Table 1: The mean HCHO column bias (relative bias) of GEOS-Chem simulations under the normal condition and drought condition in South Korean region.

| Unit: $10^{16}$ molec. cm$^{-2}$ | Standard GEOS-Chem | WD | JD |
|---|---|---|---|
| Normal | 0.22 (19.82 %) | 0.18 (16.22 %) | 0.13 (11.71 %) |
| Drought | 0.42 (35.89 %) | 0.36 (30.77 %) | 0.26 (22.22 %) |

Line 195: I assume monoterpenes and sesquiterpenes are grouped into the lumped alkenes. As indicated in Figure S3, isoprene emissions are far higher than those of other terpenoids. Could you provide more vegetation information (e.g., broadleaf and conifer tree fractions) to explain this?

→ We updated the original Figure S3 to Figure S6 in the revised manuscript to include monoterpenes and sesquiterpenes (Figure below). As shown in Figure S6, isoprene emissions are much higher than monoterpenes and sesquiterpenes, which is consistent with the previous study (Jang et al., 2020; Kim et al., 2015). According to the Korea Forest Service (https://english.forest.go.kr/kfsweb/kfi/kfs/cms/cmsView.do?cmsId=FC_001679&mn=UENG_01_03#:~:text=Status%20of%20Forest%20in%20Korea&text=The%20forested%20area%20in%20Korea,times%20higher%20than%20in%201953), about 60% of forests in Korea consist of deciduous-leaved forests and mixed forests, and 36.9% are coniferous forests. As isoprene emissions are associated with broadleaf trees and monoterpene emissions are associated with conifer trees, higher isoprene emissions can be explained by these forest fractions. Also, oak trees (*Quercus mongolica, Quercus variabilis, and Quercus acutissima*), which are known to be major sources of high isoprene emissions, are dominant species in the deciduous-leaved forests (Lee et al., 2025) so this can explain high isoprene emission in South Korea. Responding to the reviewer's comment, we added the following sentences to the revised manuscript.

"The reason why isoprene emissions are higher than monoterpenes and sesquiterpenes is because of the type of forests in South Korea. According to the Korea Forest Service (https://english.forest.go.kr/kfsweb/kfi/kfs/cms/cmsView.do?cmsId=FC_001679&mn=UENG_01_03#:~:text=Status%20of%20Forest%20in%20Korea&text=The%20forested%20area%20in%20Korea,times%20higher%20than%20in%201953), about 60% of forests in Korea consist of deciduous-leaved forests (*Quercus mongolica, Quercus variabilis, and Quercus acutissima*) and mixed forests, and 36.9% are coniferous forests. The deciduous-leaved trees are well-known sources of biogenic isoprene."

[Figure]

Figure S6: The total amounts of each BVOC emission under the normal and the drought conditions in South Korea in the standard GEOS-Chem. The each BVOCs emission include isoprene, MTPA (monoterpenes including α-pinene, β-pinene, sabinene, and carene), MTPO (monoterpenes including myrcene, ocimene, and other monoterpenes), acetone, sesquiterpene (farnesene, β-caryophyllene, and other sesquiterpene), acetaldehyde, and lumped alkene (≥ C3).

Figure 6/7 and ozone/PM2.5 validation: The comparison here is quite generic and lacks details. The authors compared the model with the in-situ measurements. The only comparison presented is the Mean Bias (difference in mean values). I think some scatter plots of the model and observations on a weekly scale could be useful to understand the change in model performance after improving the model emissions using the OMI satellite data. Besides the common statistical metrics like $R^2$, RMSE, ME, and MB, a significance test should be conducted to determine if the improvement in emissions is statistically significant.

→ The reviewer's point is well taken. We revised Figures 6/7 to show the new scatter plot between the model and the observation (Figure 6 below). Also, we added statistical metrics such as $R^2$, RMSE, NMB in each panel in the new Figure 6. We also revised the manuscript with the new Figure 6 as below:

[Figure]

Figure 6. The scatter plot for the observed $O_3$ and the simulated $O_3$ under (a) the normal and (b) drought conditions. Black and red dots denote standard GEOS-Chem and IFDMB, respectively. Normalized mean bias (NMB), coefficient of determination ($R^2$), and root mean square error (RMSE) for the standard GEOS-Chem (black) and IFDMB (red) are presented in each panel. The gray dotted line is 1:1 line. (c-d) Same as a-b but for $PM_{2.5}$.

"Figures 6a-b show scatter plots of daytime (7am – 6pm) mean $O_3$ concentrations between the surface observations and the model outputs (black for standard GEOS-Chem and red for IFDMB) under the normal condition and the drought condition. Under the normal condition (Fig. 6a), the mean $O_3$ concentrations in South Korea were 35.50 ppbv and 46.90 ppbv in the observation and the standard GEOS-Chem, respectively. The standard GEOS-Chem had positive $O_3$ biases in most of the measurement sites, which was indicated by the normalized mean biases (NMB) of 33.58 %. After using posterior isoprene emissions estimated by the IFDMB, the modeled $O_3$ concentrations decreased in most of the South Korean region. The mean $O_3$ concentrations in the IFDMB were 44.23 ppbv under the normal condition, indicating that the mean $O_3$ concentrations were reduced by 2.66 ppbv (5.67 %) with respect to the standard GEOS-Chem by applying IFDMB. As a result, the NMB in IFDMB was 25.91 % under the normal condition, which was reduced by 7.67 % compared to the standard GEOS-Chem. Other metrics such as coefficient of determination ($R^2$) and root mean square error (RMSE) also show improvement in IFDMB compared to the standard GEOS-Chem (Fig. 6a). Under the drought condition (Fig. 6b), the mean observed $O_3$ concentrations in South Korea was 43.15 ppbv, which was higher than those under the normal condition. The increase in $O_3$ concentrations under the drought condition was consistent with the expectation of a VOC-limited regime in response to increasing HCHO yet no change in $NO_2$ under the drought condition as seen by OMI (Fig. S7d). The mean $O_3$ concentrations in the standard GEOS-Chem

was 55.42 ppbv under the drought condition with the NMB of 31.04 %. In IFDMB, the mean $O_3$ concentrations was 50.47 ppbv under the drought condition, which was reduced by 4.95 ppbv (8.93 %) with respect to the standard GEOS-Chem. This is consistent with the VOC-limited regime where the reduction in isoprene emissions by the IFDMB leads to reduced ozone concentrations. The NMB in the IFDMB was 19.63 % under the drought condition, reduced by 11.41 % compared to the standard GEOS-Chem. As under the normal condition, the improvement was also indicated by higher $R^2$ and lower RMSE. Thus, the modeled $O_3$ concentration was found to be improved by applying the IFDMB method for better isoprene emissions modeling.

Figures 6c-d show scatter plots of daytime (7am – 6pm) mean $PM_{2.5}$ concentrations between the surface observations and the model outputs (black for standard GEOS-Chem and red for IFDMB) under the normal condition and the drought condition. Under the normal condition (Fig. 6c), the mean $PM_{2.5}$ concentrations were 16.41 $\mu g/m^3$ and 12.42 $\mu g/m^3$ in the observation and the standard GEOS-Chem, respectively. The standard GEOS-Chem had negative $PM_{2.5}$ biases in most of the measurement sites except for the Seoul metropolitan area. The NMB for the standard GEOS-Chem was -20.09 %. In IFDMB, contrary to ozone, the changes in $PM_{2.5}$ concentration were not significant as indicated by the NMB, $R^2$, and RMSE. Such insignificant changes in $PM_{2.5}$ in IFDMB were also found under the drought condition (Fig. 6d).”

Equation 7 and Figure 8. The analysis here is confusing. I think the authors are arguing that drought stress is the main driver of the isoprene emission bias. However, the analysis focuses on temperature. Although high temperatures often coincide with drought in many cases, there are two drivers of vegetation water stress: one is the high Vapor Pressure Deficit (VPD) caused by a dry and hot atmosphere, and the other is dry soil conditions, which determine the water supply for plants. Additionally, long-lasting droughts are mainly controlled by a lack of water. However, the equation and analysis here use the soil moisture parameter (βt) as the indicator of drought severity but use temperature as the input for addressing the isoprene emission bias. This raises the question: is the bias caused by drought, or temperature, or both?

→ We agree with the reviewer's point. We think that the isoprene emission biases were caused by both drought and temperature. First, the standard GEOS-Chem does not have the ecophysiology module, which means it cannot simulate soil moisture parameter ($\beta_t$) and thus the impact of the drought stress on the isoprene emissions. That's the main reason why the standard GEOS-Chem has significant biases in the drought conditions. In addition, we found that the isoprene emissions in standard GEOS-Chem were overestimated mainly in high temperatures in both normal and drought conditions (Fig. 7a). This is because the isoprene emissions in the standard GEOS-Chem have a factor for temperature (Section 2.4). Given these, we chose to use soil moisture parameter ($\beta_t$) to separate the normal and drought conditions in the model and then use surface temperature to adjust the model emissions in each condition. Responding to the reviewer's comment, we added the following clarification in the main text:

“This suggests that MEGAN2.1 implemented in the standard GEOS-Chem tends to overestimate isoprene emissions compared to those in the IFDMB in high surface temperatures under both normal and drought conditions. The isoprene emissions in the standard GEOS-Chem have a strong dependence on temperature (Section 2.4). This dependence may be overestimated in South Korea under high temperature conditions.

Given that a lack of ecophysiology module to simulate soil moisture (βt) was the main reason why the standard GEOS-Chem has significant biases in the drought condition, we constructed

an equation to adjust MEGAN2.1 isoprene emissions by using soil moisture parameter (βt) to separate the normal and drought conditions in the model and using surface temperature to adjust MEGAN2.1 isoprene emissions in each condition."

**Reference**

De Smedt, I., Pinardi, G., Vigouroux, C., Compernolle, S., Bais, A., Benavent, N., Boersma, F., Chan, K. L., Donner, S., Eichmann, K. U., Hedelt, P., Hendrick, F., Irie, H., Kumar, V., Lambert, J. C., Langerock, B., Lerot, C., Liu, C., Loyola, D., Piters, A., Richter, A., Rivera Cárdenas, C., Romahn, F., Ryan, R. G., Sinha, V., Theys, N., Vlietinck, J., Wagner, T., Wang, T., Yu, H., and Van Roozendael, M.: Comparative assessment of TROPOMI and OMI formaldehyde observations and validation against MAX-DOAS network column measurements, Atmos. Chem. Phys., 21, 12561-12593, 10.5194/acp-21-12561-2021, 2021.

Müller, J. F., Stavrakou, T., Oomen, G. M., Opacka, B., De Smedt, I., Guenther, A., Vigouroux, C., Langerock, B., Aquino, C. A. B., Grutter, M., Hannigan, J., Hase, F., Kivi, R., Lutsch, E., Mahieu, E., Makarova, M., Metzger, J. M., Morino, I., Murata, I., Nagahama, T., Notholt, J., Ortega, I., Palm, M., Röhling, A., Stremme, W., Strong, K., Sussmann, R., Té, Y., and Fried, A.: Bias correction of OMI HCHO columns based on FTIR and aircraft measurements and impact on top-down emission estimates, Atmos. Chem. Phys., 24, 2207-2237, 10.5194/acp-24-2207-2024, 2024.

Potosnak, M. J., LeStourgeon, L., Pallardy, S. G., Hosman, K. P., Gu, L., Karl, T., Geron, C., and Guenther, A. B.: Observed and modeled ecosystem isoprene fluxes from an oak-dominated temperate forest and the influence of drought stress, Atmospheric Environment, 84, 314-322, https://doi.org/10.1016/j.atmosenv.2013.11.055, 2014.

Seco, R., Karl, T., Guenther, A., Hosman, K. P., Pallardy, S. G., Gu, L., Geron, C., Harley, P., and Kim, S.: Ecosystem-scale volatile organic compound fluxes during an extreme drought in a broadleaf temperate forest of the Missouri Ozarks (central USA), Global Change Biology, 21, 3657-3674, doi:10.1111/gcb.12980, 2015.

Seco, R., Holst, T., Davie-Martin, C. L., Simin, T., Guenther, A., Pirk, N., Rinne, J., and Rinnan, R.: Strong isoprene emission response to temperature in tundra vegetation, Proceedings of the National Academy of Sciences, 119, e2118014119, 10.1073/pnas.2118014119, 2022. Wang, H., Lu, X., Seco, R., Stavrakou, T., Karl, T., Jiang, X., Gu, L., and Guenther, A. B.: Modeling Isoprene Emission Response to Drought and Heatwaves Within MEGAN Using Evapotranspiration Data and by Coupling With the Community Land Model, Journal of Advances in Modeling Earth Systems, 14, e2022MS003174, https://doi.org/10.1029/2022MS003174, 2022.

Zhu, L., González Abad, G., Nowlan, C. R., Chan Miller, C., Chance, K., Apel, E. C., DiGangi, J. P., Fried, A., Hanisco, T. F., Hornbrook, R. S., Hu, L., Kaiser, J., Keutsch, F. N., Permar, W., St. Clair, J. M., and Wolfe, G. M.: Validation of satellite formaldehyde (HCHO) retrievals using observations from 12 aircraft campaigns, Atmos. Chem. Phys., 20, 12329-12345, 10.5194/acp-20-12329-2020, 2020.

---

## Author Comment (AC3)

**Review of "Modeling the Drought Stress Impact on Summertime Biogenic Isoprene Emissions in South Korea" by Jeong et al.**

This manuscript by Jeong et al. examines the impact of drought stress on biogenic isoprene emissions in South Korea using satellite formaldehyde (HCHO) data and the GEOS-Chem model with MEGAN2.1. The authors found that while OMI satellite data showed a 5.4% increase in HCHO under drought conditions, the model predicted a much higher 20.23% increase, indicating an overestimation of isoprene emissions. When the authors tested existing drought stress algorithms—originally developed for the Southeastern United States—they failed to correct this overestimation. To address this issue, they applied an Iterative Finite Difference Mass Balance (IFDMB) method, which reduced isoprene emissions by 60% under drought conditions and brought the modeled HCHO increase (10.71%) closer to observations. Finally, they proposed an empirical equation for adjusting isoprene emissions in South Korea as a function of surface temperature. Overall, this manuscript is well-written and well-organized, aligning with the journal's scope. However, I have concerns regarding certain aspects of the methodology. I recommend the manuscript for publication after substantial revisions.

→ We sincerely appreciate the reviewer who gave the constructive comments to improve the manuscript. Their comments are reproduced below followed by our responses in blue. The corresponding edits in the manuscript are highlighted with red color.

**Research Scope**

• The true goal of the study is unclear. While the authors claim to examine the impact of drought stress, they primarily test existing drought stress algorithms without conducting an in-depth investigation into the actual effects of drought stress. The improved results stem from the IFDMB method, suggesting that the study may be more focused on an observation-constrained emissions inversion application rather than the direct impact of drought stress. At this point, it is unclear whether the model bias is due to drought stress, inherent issues in the model algorithm, or biases in the model input (e.g., temperature). Additionally, discrepancies appear to exist even under normal conditions.

→ The true goal of this study is to improve the simulation of isoprene emissions under the drought conditions in South Korea. As presented in the new Table 1 in the revised manuscript, the biases of the mean HCHO column in the standard GEOS-Chem increased by 16.07 % under the drought conditions compared to the normal conditions. In addition, the spatial correlation between the HCHO columns from OMI and the standard GEOS-Chem decreased under the drought conditions compared to the normal conditions (please refer to the response to the reviewer's comment below). The main reason for the worsening performance of GEOS-Chem under drought conditions was that the standard GEOS-Chem did not have the ecophysiology module to simulate the soil parameter ($\beta_t$ in Section 2.4), and thus it cannot simulate the impact of drought stress on the isoprene emissions. As two existing drought stress algorithms for GEOS-Chem were found to be ineffective in South Korea (Table 1), we estimated isoprene emissions by using the IFDMB method and provided the empirical equations to improve the simulation of isoprene emissions. Therefore, we believe that the IFDMB was used as a tool to achieve the true goal of this study: the improvement of the simulation of isoprene emissions in GEOS-Chem under the drought conditions in South Korea.

Table 1: The mean HCHO column bias (relative bias) of GEOS-Chem simulations under the normal condition and drought condition in South Korean region.

| Unit: $10^{16}$ molec. cm$^{-2}$ | Standard GEOS-Chem | WD | JD |
|---|---|---|---|
| Normal | 0.22 (19.82 %) | 0.18 (16.22 %) | 0.13 (11.71 %) |
| Drought | 0.42 (35.89 %) | 0.36 (30.77 %) | 0.26 (22.22 %) |

**Model & Data Quality Validation**

• Basic model performance evaluation is required. Since the main objective of this study is to assess the impact of drought stress on biogenic emissions estimation, the model must demonstrate reasonable performance in terms of intensity and spatial distribution. Without this, it is difficult to determine the true source of uncertainty.

→ We agreed with the reviewer. The model we used was substantially evaluated with the KORUS-AQ field campaign data as stated in the paper (Park et al., 2021). In the new Figure S5a, we showed the scatter plot between the OMI HCHO column and the GEOS-Chem HCHO column under the normal conditions in South Korea. Although GEOS-Chem HCHO column showed some biases in intensity, the slope for the regression line was 0.90 and the correlation between the two was 0.58 (statistically significant at a 99% confidence level based on the Student's t test). It indicates that GEOS-Chem has reasonable performance in terms of the HCHO spatial distribution. However, the correlation was 0.32 under the drought conditions (the new Fig. S5b), indicating that the worsening performance of GEOS-Chem under the drought conditions. We added the following sentences in the revised manuscript.

"The spatial correlation between the OMI HCHO column and the standard GEOS-Chem under the normal condition was 0.58 (Fig. S5a), which was statistically significant at a 99% confidence level based on the Student's *t*-test. It indicates that the GEOS-Chem has reasonable performance in terms of spatial distribution of the HCHO column under the normal condition. However, the lower spatial correlation between two (0.32) was found under the drought condition (Fig. S5b), which consistently indicated the worsening performance of GEOS-Chem under the drought condition."

[Figure]

Figure S5: (a) Scatterplot of the OMI HCHO column and the GEOS-Chem HCHO column under the normal conditions in South Korea. Each dot denotes HCHO column value at each grid point in South Korea. The gray dashed line denotes 1:1 line and the red line denotes linear regression line. The slope for the regression line is shown at the right side of the panel with the correlation coefficient (Corr.) between two HCHO columns. (b) Same as a but for the drought conditions.

• The quality of the dataset should be carefully examined. There are several discrepancies that are difficult to understand. In Figures 2a and 2b, OMI HCHO shows a slight increase from normal to drought conditions. However, there is a notable decrease over the Taebaek Mountains, which raises concerns. Can the authors explain this? Otherwise, this may indicate potential quality issues in the satellite data or the drought-day selection process.

→ As stated in the manuscript, the Level 3 OMI HCHO dataset used in this study (OMHCHOd) is the dataset in which bad HCHO retrievals are already filtered out. As this dataset has been widely used in other studies (e.g., Wasti and Wang, 2022), we believe that there are no quality issues in the satellite data. The reason why there was a notable HCHO decrease over the Taebaek Mountains under the drought conditions (all category combined) might be that the HCHO over the Taebaek Mountains increased only under the extreme drought category as shown in the new Figure S4 in the revised manuscript. We added the following sentences in the revised manuscript:

"For example, the OMI HCHO column over the northeastern parts of South Korea (Taebaek Mountains), which showed a decrease under the drought condition (Figs. 2a-c), showed an increase only under the extreme drought category (Fig. S4)."

• Separation of normal and drought conditions is not clearly defined. Was this classification determined on a weekly basis, or was it based on specific drought years (e.g., 2016, 2017, and 2018)? If the latter, what years were used as the baseline for normal conditions?

→ We defined the normal and drought conditions on a weekly basis. As shown in Figure S1, the normal weeks and drought weeks could be defined in each summer (Dot denotes drought week). This calculation was done in every grid cell based on the DEDI at the corresponding

grid cell. Given this, all analyses in this study were conducted on a weekly basis. To clarify, we added the following sentences in the manuscript:

"For each grid point, therefore, the normal (DEDI > -0.49) and drought (DEDI ≤ -0.49) conditions were defined based on a weekly basis using DEDI during three summers (2016 – 2018)."

**IFDMB Method**

• The application of IFDMB should be reviewed more carefully. In this study, the authors attempt to adjust biogenic isoprene emissions using OMI HCHO column density. However, unlike primary pollutants, HCHO is a secondary pollutant formed through the oxidation of VOCs. In the current model simulation, the contribution of anthropogenic VOC precursors is as significant as that of biogenic isoprene, as evidenced by the simulated HCHO spatial distribution (compare Figures 2d and 3a). The spatial distributions of HCHO from the model and OMI do not appear to be consistent, which warrants further review. As the authors stated in the manuscript, the IFDMB method does not account for the spatial transport of precursors. If the locations of emissions do not align with the locations of high HCHO concentrations, how can this method be justified?

→ This comment is associated with the reviewer's comment below. Please refer to our response to the reviewer's comment below.

• Provide the spatial distribution of anthropogenic and biogenic VOC (or isoprene) emissions from the model. The authors need to justify the application of the IFDMB method, which uses observed HCHO to adjust biogenic isoprene emissions exclusively.

→ The spatial distributions of anthropogenic VOC (AVOC) are presented in the new Figure S8 and those of biogenic isoprene emissions are presented in the original Figures 3b-c. The AVOC emissions were localized over northwestern and southeastern parts of South Korea where major metropolitan areas are located while isoprene emissions were distributed over most of South Korea. Also, AVOC emissions were consistent throughout the normal and drought conditions (Fig. S8 and Fig. 3a), which means that changes in HCHO under the drought conditions were caused by the changes in isoprene emissions. However, the assumption behind the application of IFDMB method is that anthropogenic VOC emissions used in this study are correct, at least with much higher accuracy than isoprene emissions, which is the main caveat in this study. However, the impact of AVOC emissions may be localized and spatially distinct from that of isoprene emissions. We added the following sentences in the revised manuscript.

"The assumption behind the application of IFDMB method was that AVOC emissions used in this study were correct, at least with much higher accuracy than isoprene emissions, which is the main caveat in this study. However, AVOC emissions were localized over the northwestern and southeastern parts of South Korea where major metropolitan areas are located (Fig. S8) while isoprene emissions were distributed over most of South Korea (Figs. 3b-c). Also, AVOC emissions were consistent throughout the normal and drought conditions (Fig. S8 and Fig. 3a), which means that changes in HCHO under the drought condition were caused by the changes in isoprene emissions. Therefore, the impact of AVOC emissions may be localized and spatially distinct from that of isoprene emissions."

[Figure]

Figure S8: The spatial distribution of anthropogenic VOC (AVOC) emissions under normal condition (left) and drought condition (right).

• Figures 6 and 7 should be updated. These figures are confusing and misleading, as they present concentration and bias together. Please provide separate panels for concentrations (model and observations) and biases.

➔ We have changed the original Figures 6/7 to the new scatter plot between the model and the observation (Figure 6 below). Also, we added statistical metrics such as $R^2$, RMSE, NMB in each panel in the new Figure 6. We also revised the manuscript with the new Figure 6 as below:

[Figure]

Figure 6. The scatter plot for the observed $O_3$ and the simulated $O_3$ under (a) the normal and (b) drought conditions. Black and red dots denote standard GEOS-Chem and IFDMB, respectively. Normalized mean bias (NMB), coefficient of determination ($R^2$), and root mean square error (RMSE) for the standard GEOS-Chem (black) and IFDMB (red) are presented in each panel. The gray dotted line is 1:1 line. (c-d) Same as a-b but for $PM_{2.5}$.

"Figures 6a-b show scatter plots of daytime (7am – 6pm) mean $O_3$ concentrations between the surface observations and the model outputs (black for standard GEOS-Chem and red for IFDMB) under the normal condition and the drought condition. Under the normal condition (Fig. 6a), the mean $O_3$ concentrations in South Korea were 35.50 ppbv and 46.90 ppbv in the observation and the standard GEOS-Chem, respectively. The standard GEOS-Chem had positive $O_3$ biases in most of the measurement sites, which was indicated by the normalized mean biases (NMB) of 33.58 %. After using posterior isoprene emissions estimated by the IFDMB, the modeled $O_3$ concentrations decreased in most of the South Korean region. The mean $O_3$ concentrations in the IFDMB were 44.23 ppbv under the normal condition, indicating that the mean $O_3$ concentrations were reduced by 2.66 ppbv (5.67 %) with respect to the standard GEOS-Chem by applying IFDMB. As a result, the NMB in IFDMB was 25.91 % under the normal condition, which was reduced by 7.67 % compared to the standard GEOS-Chem. Other metrics such as coefficient of determination ($R^2$) and root mean square error (RMSE) also show improvement in IFDMB compared to the standard GEOS-Chem (Fig. 6a). Under the drought condition (Fig. 6b), the mean observed $O_3$ concentrations in South Korea was 43.15 ppbv, which was higher than those under the normal condition. The increase in $O_3$ concentrations under the drought condition was consistent with the expectation of a VOC-limited regime in response to increasing HCHO yet no change in $NO_2$ under the drought condition as seen by OMI (Fig. S7d). The mean $O_3$ concentrations in the standard GEOS-Chem was 55.42 ppbv under the drought condition with the NMB of 31.04 %. In IFDMB, the mean $O_3$ concentrations was 50.47 ppbv under the drought condition, which was reduced by 4.95 ppbv (8.93 %) with respect to the standard GEOS-Chem. This is consistent with the VOC-limited regime where the reduction in isoprene emissions by the IFDMB leads to reduced ozone concentrations. The NMB in the IFDMB was 19.63 % under the drought condition, reduced by 11.41 % compared to the standard GEOS-Chem. As under the normal condition, the improvement was also indicated by higher $R^2$ and lower RMSE. Thus, the modeled $O_3$ concentration was found to be improved by applying the IFDMB method for better isoprene emissions modeling.

Figures 6c-d show scatter plots of daytime (7am – 6pm) mean $PM_{2.5}$ concentrations between the surface observations and the model outputs (black for standard GEOS-Chem and red for IFDMB) under the normal condition and the drought condition. Under the normal condition (Fig. 6c), the mean $PM_{2.5}$ concentrations were 16.41 µg/m$^3$ and 12.42 µg/m$^3$ in the observation and the standard GEOS-Chem, respectively. The standard GEOS-Chem had negative $PM_{2.5}$ biases in most of the measurement sites except for the Seoul metropolitan area. The NMB for the standard GEOS-Chem was -20.09 %. In IFDMB, contrary to ozone, the changes in $PM_{2.5}$ concentration were not significant as indicated by the NMB, $R^2$, and RMSE. Such insignificant changes in $PM_{2.5}$ in IFDMB were also found under the drought condition (Fig. 6d)."

• Clarify the data processing in Figure 8a–d. Please elaborate on how the data were processed and explain what each point represents. If possible, please provide the raw data points before binning in 0.2 K intervals.

→ Thanks for the reviewer's careful comment. The original Figure 8 was revised to Figure 7 in the revised manuscript. Each dot in the Figures 7a-b represents isoprene emission value in the standard GEOS-Chem versus that in IFDMB at each grid point in South Korea. The isoprene emission value was divided by LAI value at the corresponding grid point to consider the different vegetation coverage. The surface temperature value at the corresponding grid was also overlaid at each dot. Each dot in Figures 7c-d represents $\gamma_{SM\_OMI}$ value (Eq. 6) at the corresponding surface temperature. We revised the caption for Figure 7 in the revised manuscript for a clear explanation and added the new figure (Fig. S9) to provide the raw data points before binning in 0.2 K interval in Figs. 8c-d.

[Figure]

Figure S9: $\gamma_{SM\_OMI}$ with respect to the surface temperatures in (c) the normal and (d) the drought conditions. The red dotted lines indicate the fitted lines.